# A Design Methodology for the Seismic Retrofitting of Existing Frame Structures Post-Earthquake Incident Using Nonlinear Control Systems

**Assaf Shmerling** [1],* and **Matthias Gerdts** [2]

1    Department of Civil and Environmental Engineering, Ben-Gurion University of the Negev, Beer-Sheva 84105, Israel

2    Institute of Applied Mathematics and Scientific Computing, Universität der Bundeswehr München, 85577 Neubiberg, Germany

\*    Correspondence: assafs@bgu.ac.il

**Abstract:** A structural design methodology for retrofitting weakened frame systems following earthquakes is developed and presented. The design procedure refers to frame systems in their degraded strength and stiffness states and restores their dynamic performance using nonlinear control systems. The control law associated with the employed systems regards the gains between the negative state feedback and the control force, which consists of linear, nonlinear, and hysteretic portions. Structural optimization is introduced in designing the nonlinear control systems, and the controller gains are optimized using the fixed-point iteration to improve the frame system's dynamic performance. The fixed-point iteration method relates to first-order PDE equations; hence, a new state-space formulation for weakened inelastic frame systems is developed and presented using the frame system's lateral force equilibrium equation. The design scheme and optimization strategy differ from designing passive control systems, given that the nonlinear control system's force consists of linear, nonlinear, and hysteretic portions. The utilization of the fixed-point iteration in the structural design area is by itself a novel application due to its robustness in addressing the gains of any type of nonlinear control system. This paper's nonlinear control system chosen to exhibit the application is Buckling Restrained Braces (BRBs) since force consists of linear and hysteretic portions. The implementation of hysteretic control force is rare in structural control applications. In the case of BRBs, the fixed-point iteration optimizes the cross-sectional areas. Two system optimization examples of 3-story and 15-story inelastic frames are provided and described. The examples demonstrate the fixed-point iteration's applicability and robustness in optimizing control gains of nonlinear systems and regulating the dynamic response of weakened frame structures.

**Keywords:** Inelastic frame systems; post-earthquake design; nonlinear control; fixed-point iteration; control gain optimization

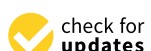

## 1. Introduction

Seismic retrofitting inelastic lateral load resisting systems is challenging primarily due to the unpredictable earthquake response and final damage state. It becomes even more challenging when retrofitting a system that is already damaged, following an earthquake incident, since the system is characterized by stiffness and strength degradations. Nevertheless, the design philosophy and application remain the same when utilizing control systems for reinstating and improving the existing system. According to well-accepted global damage indices, the weakened state of the structure after an earthquake can be considered by its maximum fundamental vibrations period, related to the most significant stiffness degradation.

The global damage indices indicate the damage state of the whole system to determine its functionality level—unlike local damage indices, which look at a particular member

(see, for example, [1–4]). Global damage equations compare the parameters of the structure (e.g., modal frequencies) before and after the earthquake excitation to quantify the system's weakening. For example, DiPasquale et al. [5] developed the "plastic softening" and "final softening" global damage indices for reinforced concrete structures. The "plastic softening" damage index provides a good measurement of plastic deformation and soil-structure interaction during seismic excitation (regarding the final and maximum period ratio). In contrast, the "final softening" indicates the state of the structure after dynamic excitation (regarding the initial and final period ratio). DiPasquale and Cakmak [6] followed suit and developed the "maximum softening" global damage index, indicating the ultimate stiffness degradation related to the maximum fundamental vibration period under seismic excitation. In recent years, a few researchers followed suit in developing new terms for the global damage index (e.g., [7,8]). According to the review paper of Williams and Sexsmith [9], it is considered the best indicator of the global damage state. In this paper, we adopt the maximum softening philosophy and address the maximum fundamental vibrations period of the frame system. At this point, we employ nonlinear control systems.

Various design methodologies have been developed and exemplified their effectiveness in regulating the earthquake response of frame systems by utilizing control systems. The procedures for designing control systems usually aim at optimizing their parameters to minimize a particular objective function as a part of an optimization problem. See, for example, the applications of Lagrange multipliers in [10–13], gradient-based optimization in [14–20], the linear quadratic regulator in [21–26], and the direct probability integral approach in [27–29]. In such cases, when optimizing static parameters (i.e., static gains), most optimization algorithms combine a technique to assess/estimate/address the inelastic response by an equivalent linear analysis and, if necessary, recalculate the gains corresponding to the inelastic system. One example is the procedure by Shmerling and Levy [18], which utilizes the direct-displacement-based approach for simplifying inelastic frame systems into an equivalent linear system in interstory drift deformations.

Researchers with unique optimization typologies usually regard the system's equilibrium equation, induced force, or stress. For example, Potra and Simiu [30] express the column's stress under extreme loads (e.g., earthquakes) and introduce it to a nonlinear dynamic programming algorithm. Shmerling and Levy [31] use the general system interconnection paradigm to represent the dynamic equilibrium of rigid frame systems in the Laplace domain. Smarra et al. [32] represent the LQR objective function and state-space in a discrete manner suitable for the predictive horizon approach implementation. This paper proposes a new inelastic state-space formulation for nonlinear static gains control systems and later refers to BRBs as a nonlinear control system following [33,34].

The effectiveness of the BRBs application in upgrading and improving the dynamic performance of civil structures is well acknowledged. Besides being considered for upgrading rigid frame systems, BRBs have also been proposed to enhance the performance of different infrastructures, such as nuclear plant turbine buildings [35] and steel arch bridges [36]. There are various optimization techniques for designing BRBs. Balling et al. [37] present a nine-step algorithm that performs nonlinear time history and reconfigures the BRBs according to the redesign equations. Hoffman and Richards [38] introduce four different genetic algorithms (baseline, forced diversity, adaptive mutation, and noncrossover adaptive mutation). Abedini et al. [39] solve an optimization problem in which the objective function is the BRBs' weight and the dissipated energy using the metaheuristic salp swarm and colliding bodies algorithms. Pan et al. [40] design procedure that calculates the minimal weight of BRBs subject to the global buckling prevention criterion and the stiffness–strength relationship curve. Rezazadeha and Talatahari [41] address the seismic input energy to the structure and the absorbed yielding energy and utilize the vibrating particles system algorithm to achieve the optimal BRBs configuration. Tu et al. [42] optimally allocate BRBs to frame structures while referring to the deformation and damage constraints.

In this paper, the motivation for employing BRBs stems from its control law, which consists of linear and hysteretic control forces corresponding to the resisting force's elastic

and inelastic portions. Control force of hysteretic nature is more challenging to implement in control theory due to the gains being subject to integration terms. Nevertheless, the fixed-point iteration method is suitable for such a control system due to addressing its state trajectory.

## 2. Inelastic State-Space

The inelastic state-space formulation presented herein enhances the equations scheme in [26] for frame structures, which refers to the lateral force equilibrium of an inelastic rigid frame system under lateral loads. In this paper, the frame structure is equipped with a nonlinear control system of static gains that induces either nonlinear, hysteretic, or linear portions—depending on the system type. The nonlinear control system chosen in this paper is BRB, whose applied force consists of hysteretic and linear parts. In this case, the static gains are the BRBs' cross-sectional areas.

The addressed lateral force equilibrium considers the inelastic behavior of the structure since even while we use a control device to keep the system close to its elastic range, this may not always be the case. The expression of the condensate equation governing the lateral force equilibrium is:

$$
\begin{aligned}
&\mathbf{m}\,\ddot{\mathbf{x}}(t) + \mathbf{c}\,\dot{\mathbf{x}}(t) + \mathbf{T}'_{x\rightarrow d}\mathbf{f}^{F}\left(\dot{\mathbf{f}}^{F},\mathbf{d},\dot{\mathbf{d}}\right) + \mathbf{u}\left(\Sigma\mathbf{A},\dot{\mathbf{u}},\mathbf{x},\dot{\mathbf{x}}\right) = \mathbf{p}(t)\\
&\text{and :}\\
&\mathbf{x}(0) = 0\\
&\dot{\mathbf{x}}(0) = 0
\end{aligned}
\tag{1}
$$

where $(\ )'$ denotes the conjugate-transpose, $\mathbf{x}(t)$ is the $N$-dimensional ceilings' relative-to-ground displacement vector, $\dot{\mathbf{x}}(t)$ is the $N$-dimensional ceilings' relative-to-ground velocities vector, $\ddot{\mathbf{x}}(t)$ is the $N$-dimensional ceilings' relative-to-ground accelerations vector, $\mathbf{m}$ is the $N \times N$ static-condensate diagonal mass matrix, $\mathbf{c}$ is the $N \times N$ inherent damping matrix, $\mathbf{p}(t)$ is the $N$-dimensional lateral dynamic load vector, $\mathbf{f}^{F}\left(\dot{\mathbf{f}}^{F},\mathbf{d},\dot{\mathbf{d}}\right)$ is the $N$-dimensional structural rigid frame system's lateral resisting force vector, and $\mathbf{u}\left(\Sigma\mathbf{A},\mathbf{u},\mathbf{x},\dot{\mathbf{x}}\right)$ is the $N$-dimensional nonlinear static gains control force vector. The vector function $\mathbf{d}(t)$, which is referred by $\mathbf{f}^{F}\left(\dot{\mathbf{f}}^{F},\mathbf{d},\dot{\mathbf{d}}\right)$, is the $N$-dimensional interstory drifts vector—given by the transformation from displacement coordinates into drift coordinates:

$$
\mathbf{d}(t) = \mathbf{T}_{x\rightarrow d}\mathbf{x}(t) \ \leftrightarrow \ \mathbf{T}_{x\rightarrow d} =
\begin{bmatrix}
1 & & & \\
-1 & 1 & & \\
& \ddots & \ddots & \\
& & -1 & 1
\end{bmatrix}
\tag{2}
$$

where $\mathbf{T}_{x\rightarrow d}$ denotes the transformation matrix, and its reverse form is:

$$
\mathbf{x}(t) = \mathbf{T}_{d\rightarrow x}\mathbf{d}(t) \ \leftrightarrow \ \mathbf{T}_{d\rightarrow x} =
\begin{bmatrix}
1 & & \\
\vdots & \ddots & \\
1 & \cdots & 1
\end{bmatrix}
\tag{3}
$$

Figure 1 depicts the structural deformations in terms of $\mathbf{x}(t)$ and $\mathbf{d}(t)$ to exemplify the difference between the two.

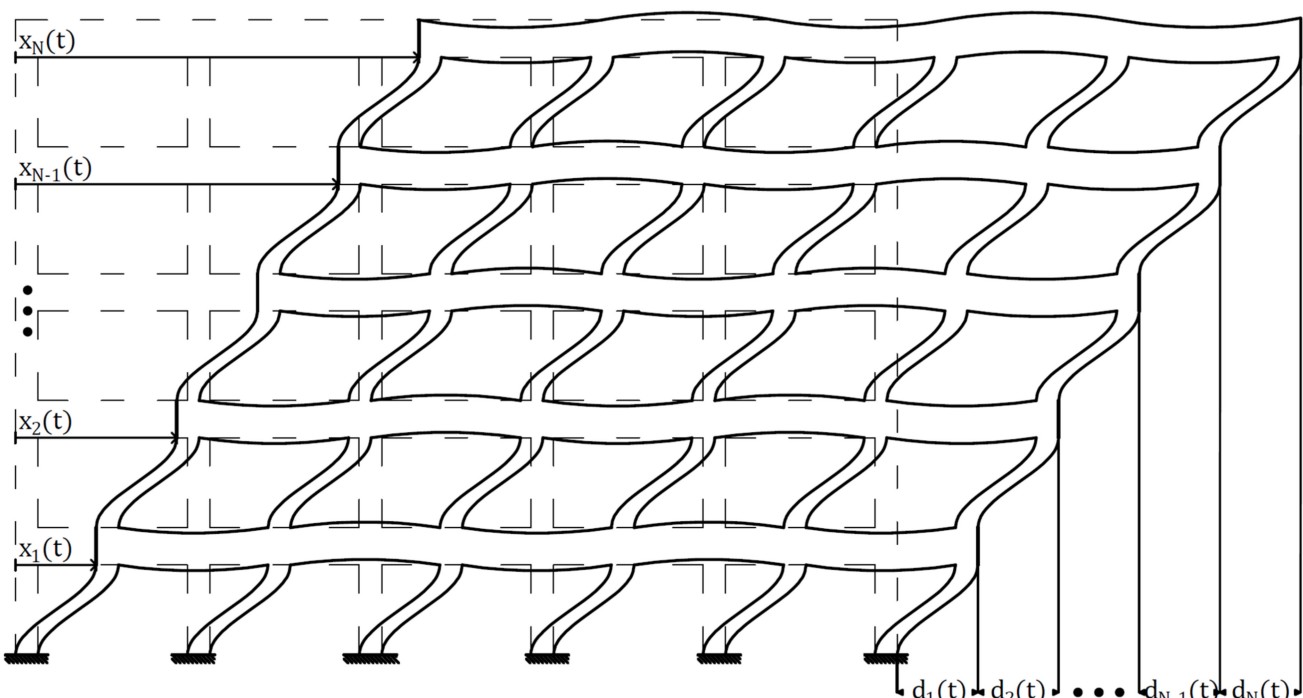

**Figure 1.** Rigid frame system deformations in terms of relative-to-ground displacements $(x_1, x_2, \ldots, x_{N-1}, x_N)$ and interstory drifts $(d_1, d_2, \ldots, d_{N-1}, d_N)$.

The hysteretic model of $\mathbf{f}^F$ is expressed as a combination of its elastic and hysteretic portions. Considering that:

$$\mathbf{f}^F\left(\dot{\mathbf{f}}^F, \mathbf{d}, \dot{\mathbf{d}}\right) = \int_0^t \dot{\mathbf{f}}^F\left(\mathbf{f}^F, \mathbf{d}, \dot{\mathbf{d}}\right)d\tau = \mathbf{k}^{F,el}\,\mathbf{d}(t) + \int_0^t \dot{\mathbf{f}}^{F,hys}\left(\mathbf{f}^F, \mathbf{d}, \dot{\mathbf{d}}\right)d\tau \tag{4}$$

where $\mathbf{k}^{F,el}$ is the $N \times N$ static-condensate matrix elastic stiffness portion about $\mathbf{d}(t)$, and $\mathbf{f}^{F,hys}$ is the $N$-dimensional hysteretic portion of $\mathbf{f}^F$, and give that:

$$\dot{\mathbf{f}}^{F,hys}(t) = \mathbf{k}^{F,hys}\left(\mathbf{f}^F, \mathbf{d}, \dot{\mathbf{d}}\right)\dot{\mathbf{d}}(t) \tag{5}$$

where $\mathbf{k}^{F,hys}$ is the $N \times N$ static-condensate hysteretic stiffness matrix, the force $\mathbf{f}^F$ is formulated as:

$$\mathbf{f}^F\left(\dot{\mathbf{f}}^F, \mathbf{d}, \dot{\mathbf{d}}\right) = \mathbf{k}^{F,el}\,\mathbf{d}(t) + \int_0^t \mathbf{k}^{F,hys}\left(\mathbf{f}^F, \mathbf{d}, \dot{\mathbf{d}}\right)\dot{\mathbf{d}}(\tau)d\tau \tag{6}$$

The control force $\mathbf{u}$ is defined by linear, hysteretic, and nonlinear control force portions. That is:

$$\mathbf{u}\left(\Sigma\mathbf{A}, \dot{\mathbf{u}}, \mathbf{x}, \dot{\mathbf{x}}\right) = \mathbf{k}^{u,el}(\Sigma\mathbf{A})\mathbf{x}(t) + \mathbf{c}^{u,el}(\Sigma\mathbf{A})\dot{\mathbf{x}}(t) + \int_0^t \mathbf{k}^{u,hys}(\Sigma\mathbf{A}, \mathbf{u}, \mathbf{x}, \dot{\mathbf{x}})\dot{\mathbf{x}}(\tau)d\tau + \mathbf{f}^{u,NL}(\Sigma\mathbf{A}, \mathbf{x}, \dot{\mathbf{x}}) \tag{7}$$

where $\Sigma\mathbf{A}$ is an $N$-dimensional vector consisting of the static gain variables, $\mathbf{k}^{u,el}$ is the $N \times N$ linear stiffness matrix, $\mathbf{c}^{u,el}$ is the $N \times N$ linear damping matrix, $\mathbf{k}^{u,hys}$ is the $N \times N$ hysteretic stiffness matrix, and $\mathbf{f}^{u,NL}$ denotes the $N$-dimensional nonlinear portion of $\mathbf{u}$.

The applied dynamic load vector $\mathbf{p}(t)$ is modeled as the quasi-static cyclic loading whose maximum amplitude reaches twice the total yielding force. As the numerical examples exemplify, adhering to such load supports the system remaining close to its elastic state under a significant earthquake to maintain our weakened frame structure.

Figure 2 illustrates the normalized nth story load $\mathbf{p}_n(t)$ about the total yielding force so that $\mathbf{f}_n^{F,yld}$ is the nth story columns' shear force at first yield versus the normalized load duration about the highest modal period $\mathbf{T}_1$.

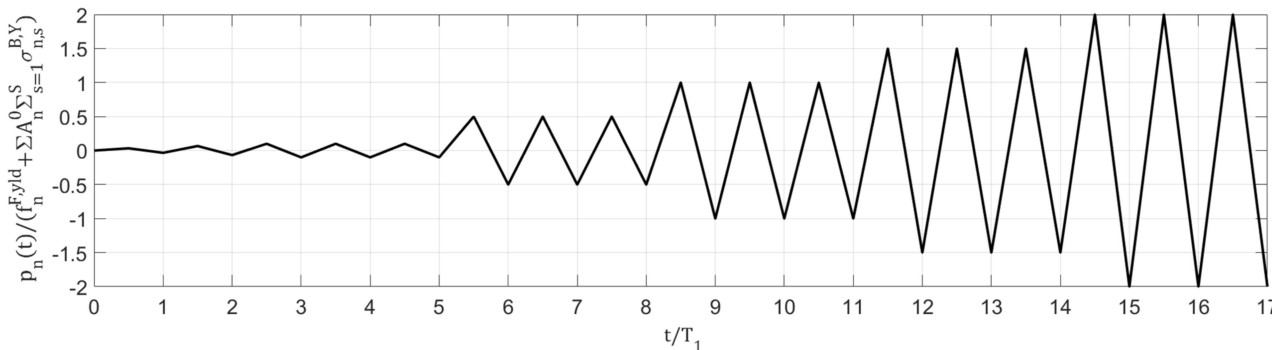

**Figure 2.** Quasi-static cyclic loading.

The inelastic state-space formulation refers to the term $\dot{\mathbf{f}}^{\mathrm{F,hys}} = \mathbf{k}^{\mathrm{F,hys}}\dot{\mathbf{d}}$ of Equation (5) as a separate entity within the following *4N*-dimensional state-vector $\mathbf{z}(t)$:

$$\mathbf{z}(t) = \begin{bmatrix} \mathbf{x}(t) \\ \int_0^t \mathbf{f}^{\mathrm{F,hys}}(\mathbf{z}(t))\mathrm{d}\tau \\ \dot{\mathbf{x}}(t) \\ \mathbf{f}^{\mathrm{F,hys}}(\mathbf{z}(t)) \end{bmatrix} \tag{8}$$

Consequently, the corresponding *4N* × *4N* state matrix $\mathbf{A}(\mathbf{z})$ is:

$$\mathbf{A}(\mathbf{z}(t)) = \begin{bmatrix} \mathbf{0} & \mathbf{0} & \mathbf{I} & \mathbf{0} \\ \mathbf{0} & \mathbf{0} & \mathbf{0} & \mathbf{I} \\ -\mathbf{m}^{-1}\mathbf{T}'_{x\to d}\mathbf{k}^{\mathrm{F,el}}\mathbf{T}_{x\to d} & \mathbf{0} & -\mathbf{m}^{-1}\mathbf{c} & -\mathbf{m}^{-1} \\ \mathbf{0} & \mathbf{0} & \mathbf{k}^{\mathrm{F,hys}}(\mathbf{z}(t))\mathbf{T}_{x\to d} & \mathbf{0} \end{bmatrix} \tag{9}$$

and the inelastic state-space equation is:

$$\dot{\mathbf{z}}(t) = \mathbf{A}(\mathbf{z}(t))\mathbf{z}(t) + \mathbf{B}\big(\mathbf{p}(t) + \mathbf{u}(\Sigma\mathbf{A}, \dot{\mathbf{u}}, \mathbf{x}, \dot{\mathbf{x}})\big) \tag{10}$$

where the *4N* × *N* input-to-state matrix $\mathbf{B}$ is:

$$\mathbf{B} = \begin{bmatrix} \mathbf{0} \\ \mathbf{0} \\ \mathbf{m}^{-1} \\ \mathbf{0} \end{bmatrix} \tag{11}$$

Since $\mathbf{z}(t)$ consists of $\mathbf{x}(t)$ and $\dot{\mathbf{x}}(t)$ the nonlinear static gains control force is henceforth denoted as $\mathbf{u}(\Sigma\mathbf{A}, \mathbf{z}(t))$.

Figure 3 depicts the closed-loop control process. The closed-loop paradigm comprises the inelastic state-space, defined above, and the control law regarding the various control force portions (linear, hysteretic, and nonlinear). The matrices $\mathbf{T}^{\mathrm{hys}}$, $\mathbf{T}^{\mathrm{lin}}$, and $\mathbf{T}^{\mathrm{NL}}$ are portion transformations matrices applied to the negative feedback of $\mathbf{z}$ to yield the respected portions of the control force, and $\mathbf{G}(\Sigma\mathbf{A}, \mathbf{z}(t))$ is the $N \times 4N$ gain matrix. The BRBs system is chosen to regulate the frame structure and to exemplify the developed methodology. The BRB's control law is composed of linear force, relating to the BRB elastic portion, and hysteretic force, relating to the material inelasticity.

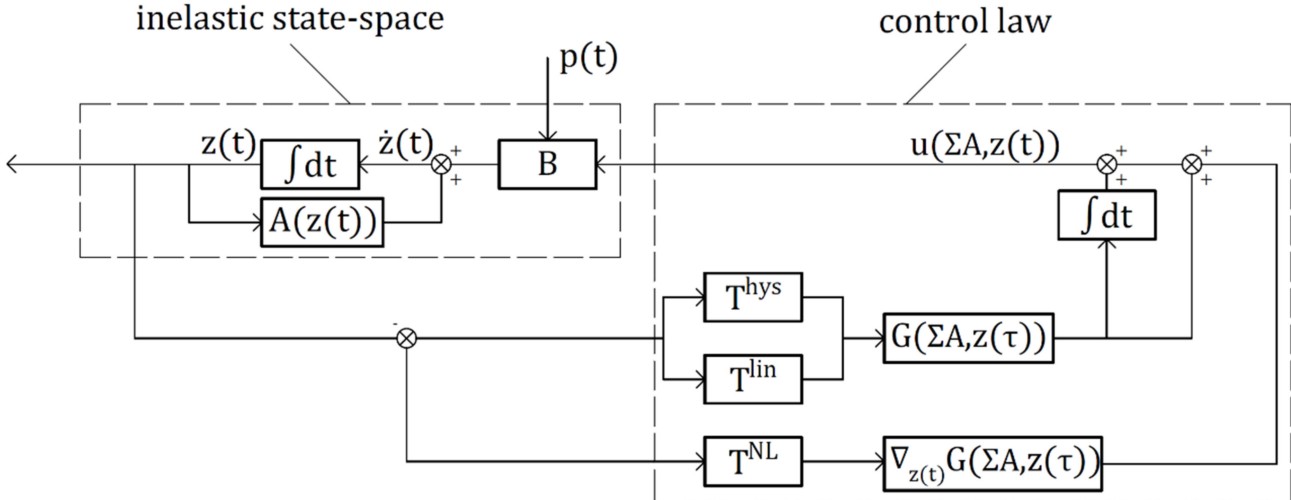

**Figure 3.** Proposed state-space paradigm.

## 3. BRB Control Law

The BRB system is chosen herein since it consists of an inelastic force portion. Inelastic behavior is more challenging to implement in control theory because of its hysteretic nature. Various research projects have examined BRB behavior over the last decades. Recently, Tremblay et al. [43] tested six BRBs segments and examined different brace cores, mechanisms, and profiles with/without unbinding material. The research shows that certain BRB specimens exhibited a predictable elastic response and a ductile and stable inelastic response, without fracture, under the cyclic quasi-static loading plus four additional large-amplitude tension cycles. This paper's analytical model considers such idealized cyclic behavior. Figure 4 depicts a rigid frame system supplied with BRBs—showcasing its installation.

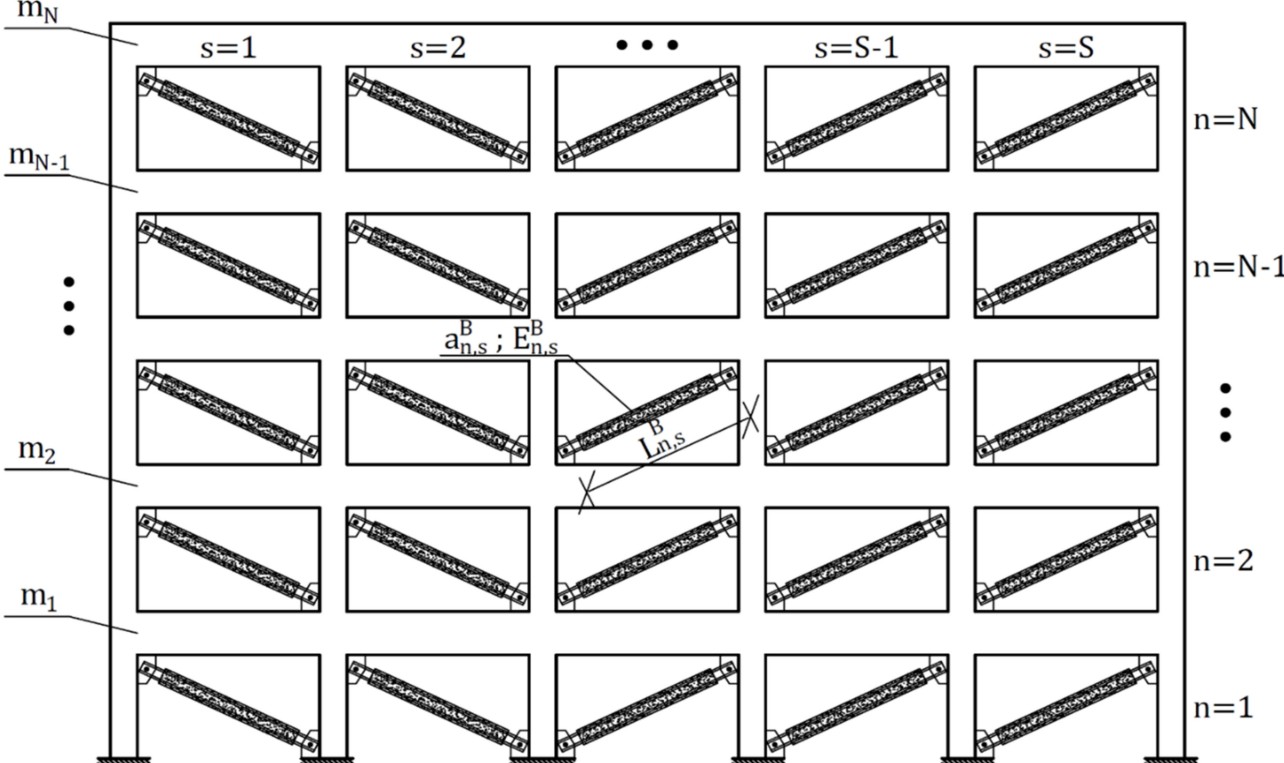

**Figure 4.** Rigid frame system elevation scheme equipped with BRBs.

The BRB cyclic model is composed of linearly elastic and hysteretic stiffness portions and refers to each span of the rigid frame system individually. Therefore, the control force of Equation (7) is reduced to:

$$\mathbf{u}\big(\mathbf{\Sigma A}, \mathbf{u}, \mathbf{x}, \dot{\mathbf{x}}\big) = \mathbf{k}^{\mathrm{u,el}}(\mathbf{\Sigma A})\mathbf{x}(t) + \int_0^t \mathbf{k}^{\mathrm{u,hys}}\big(\mathbf{\Sigma A}, \mathbf{u}, \mathbf{x}, \dot{\mathbf{x}}\big)\dot{\mathbf{x}}(\tau)\mathrm{d}\tau \tag{12}$$

and the matrices $\mathbf{k}^{\mathrm{u,el}}$ and $\mathbf{k}^{\mathrm{u,hys}}$ are expressed as:

$$\mathbf{k}^{\mathrm{u,el}}(\mathbf{\Sigma A}) = (\mathbf{T}_{\mathrm{x}\to\Delta})'\mathrm{diag}\Big\{\Big[\Big(\mathbf{a}^{\mathrm{B}}\odot\mathbf{E}^{\mathrm{B}}\oslash\mathbf{L}^{\mathrm{B}}\Big)1^{\mathrm{S}}\Big]\odot\mathbf{\Sigma A}\Big\}\mathbf{T}_{\mathrm{x}\to\Delta} \tag{13}$$

$$\mathbf{k}^{\mathrm{u,hys}}\big(\mathbf{\Sigma A}, \mathbf{u}, \mathbf{x}, \dot{\mathbf{x}}\big) = (\mathbf{T}_{\mathrm{x}\to\Delta})'\mathrm{diag}\Big\{\Big[\Big(1-\mathbf{a}^{\mathrm{B}}\Big)\odot\Big(\boldsymbol{\rho}^{\mathrm{B}}(\mathbf{\Sigma A}, \mathbf{u}, \mathbf{x}, \dot{\mathbf{x}})\odot\mathbf{E}^{\mathrm{B}}\oslash\mathbf{L}^{\mathrm{B}}\Big)1^{\mathrm{S}}\Big]\odot\mathbf{\Sigma A}\Big\}\mathbf{T}_{\mathrm{x}\to\Delta} \tag{14}$$

where $\odot$ denotes element-wise multiplication, $\oslash$ marks element-wise division, $\mathbf{a}^{\mathrm{B}}$ is an N × S matrix consisting of the ratios between the BRBs' axial plastic and elastic stiffnesses, $\mathbf{E}^{\mathrm{B}}$ is an N × S matrix composed of the BRBs' elasticity modulus, $\mathbf{A}^{\mathrm{B}}$ is an N × S matrix containing the BRBs' effective cross-sectional area, $\mathbf{L}^{\mathrm{B}}$ is an N × S matrix comprised of the BRBs' length, $\boldsymbol{\rho}^{\mathrm{B}}$ is an N × S matrix that corresponds to the material's elastic/plastic/unloading stages, $1^{\mathrm{S}}$ an S-dimensional vector of ones, $\mathbf{T}_{\mathrm{x}\to\Delta}$ is the geometric transformation matrix from $\mathbf{x}(t)$ into the BRB's axial deformation $\Delta(t)$, and $\mathbf{\Sigma A}$ is an N-dimensional vector composed of the total cross-sectional area quantities:

$$\mathbf{\Sigma A} = \begin{bmatrix}\Sigma_{s=1}^{S}A_{1,s}^{B}\\ \vdots \\ \Sigma_{s=1}^{S}A_{N,s}^{B}\end{bmatrix} \tag{15}$$

The parameter $A_{n,s}^{B}$ denotes the cross-sectional area of the BRB installed at the sth span of the nth story.

The matrix $\boldsymbol{\rho}^{\mathrm{B}}$ is defined using the Bouc–Wen model equations. Assuming the BRB does not experience stiffness and strength degradations, the unloading curve is parallel to the elastic stiffness—which gives:

$$\boldsymbol{\rho}^{\mathrm{B}}\big(\mathbf{\Sigma A}, \mathbf{u}, \mathbf{x}, \dot{\mathbf{x}}\big) = 1 - 0.5\big|\boldsymbol{\sigma}^{\mathrm{in}}\big(\boldsymbol{\sigma}^{\mathrm{B}}, \mathbf{x}\big)\oslash\big(1-\mathbf{a}^{\mathrm{B}}\big)\oslash\boldsymbol{\sigma}^{\mathrm{B,Y}}\big|^{\nu}\odot\Big[1+\mathrm{sign}\Big(\dot{\boldsymbol{\Delta}}^{\mathrm{B}}(\dot{\mathbf{x}})\odot\boldsymbol{\sigma}^{\mathrm{in}}\big(\boldsymbol{\sigma}^{\mathrm{B}}, \mathbf{x}\big)\Big)\Big]$$

and :
$$\boldsymbol{\sigma}^{\mathrm{B}}(\mathbf{\Sigma A}, \mathbf{u}) = \Big[\big(\mathbf{u}\big(\mathbf{\Sigma A}, \dot{\mathbf{u}}, \mathbf{x}, \dot{\mathbf{x}}\big)\oslash\mathbf{\Sigma A}\big)1^{S'}\Big]\odot\Big[\mathbf{A}^{\mathrm{B}}\oslash\Big(\mathbf{\Sigma A}1^{S'}\Big)\Big] \tag{16}$$

where $\boldsymbol{\sigma}^{\mathrm{B,Y}}$ is an N × S matrix of the BRBs' yield stress, $\dot{\boldsymbol{\Delta}}^{\mathrm{B}}(t)$ is the BRBs' axial deformation rate, and $\boldsymbol{\sigma}^{\mathrm{in}}(t)$ is an N × S matrix referring to the inelastic portion of the BRBs' axial stress. The matrices $\dot{\boldsymbol{\Delta}}^{\mathrm{B}}(t)$ and $\boldsymbol{\sigma}^{\mathrm{in}}(t)$ are defined as:

$$\dot{\boldsymbol{\Delta}}^{\mathrm{B}}(\dot{\mathbf{x}}) = \mathbf{T}_{\mathrm{x}\to\Delta}\dot{\mathbf{x}}(t)1^{S'} \tag{17}$$

$$\boldsymbol{\sigma}^{\mathrm{in}}\big(\boldsymbol{\sigma}^{\mathrm{B}}, \mathbf{x}\big) = \boldsymbol{\sigma}^{\mathrm{B}}\big(\mathbf{f}^{\mathrm{B}}\big) - \mathbf{a}^{\mathrm{B}}\odot\mathbf{E}^{\mathrm{B}}\oslash\mathbf{L}^{\mathrm{B}}\odot\Big(\mathbf{T}_{\mathrm{x}\to\Delta}\mathbf{x}(t)1^{S'}\Big) \tag{18}$$

The BRB cyclic model involves a control law with a linear force portion and a hysteretic force portion. That is:

$$\mathbf{u}(\mathbf{\Sigma A}, \dot{\mathbf{u}}, \mathbf{z}(t)) = -\mathbf{G}(\mathbf{\Sigma A}, \mathbf{z}(\tau))\mathbf{T}^{\mathrm{lin}}\mathbf{z}(\tau) - \int_0^t \mathbf{G}(\mathbf{\Sigma A}, \mathbf{u}, \mathbf{z}(t))\mathbf{T}^{\mathrm{hys}}\mathbf{z}(\tau)\mathrm{d}\tau$$
and :
$$\mathbf{G}(\mathbf{\Sigma A}, \mathbf{u}, \mathbf{z}(t)) = \Big[\mathbf{k}^{\mathrm{B,el}}(\mathbf{\Sigma A})\ 0\ \mathbf{k}^{\mathrm{u,hys}}(\mathbf{\Sigma A}, \mathbf{u}, \mathbf{z})\ 0\Big]$$

$$\mathbf{T}^{\mathrm{lin}} = \begin{bmatrix}\mathbf{I} & & & \\ & 0 & & \\ & & 0 & \\ & & & 0\end{bmatrix} ;\ \mathbf{T}^{\mathrm{hys}} = \begin{bmatrix}0 & & & \\ & 0 & & \\ & & \mathbf{I} & \\ & & & 0\end{bmatrix} \tag{19}$$

where $\mathbf{k}^{B,el}(\mathbf{\Sigma A})$ and $\mathbf{k}^{u,hys}(\mathbf{\Sigma A}, \mathbf{u}, \mathbf{z})$ are defined by Equations (13) and (14), respectively. The implicit nature of $\mathbf{u}$ requires an iterative optimization approach to finding the optimal $\mathbf{\Sigma A}$. Here, the fixed-point iteration is introduced since it is capable of optimizing $\mathbf{\Sigma A}$ regardless of $\mathbf{u}$ form.

## 4. Fixed-Point Iteration

The intended utilization of the fixed-point iteration optimizes the static gain variables of nonlinear nonautonomous control systems. This paper's fixed-point iteration minimizes the rigid frame system's interstory drift and the drift velocity trajectories while referring to BRBs. The objective function is subject to the inelastic state-space, initial conditions, and control variable limitations. The complete optimization problem is given by:

$$
\begin{aligned}
&\underset{\mathbf{\Sigma A}}{\text{minimize}} && \left\{ J = \int_0^{t_f} \mathbf{z}^T(t)\mathbf{Q}\mathbf{z}(t)dt = \int_0^{t_f} \mathbf{d}^T(t)\mathbf{Q}_1\mathbf{d}(t) + \dot{\mathbf{d}}^T(t)\mathbf{Q}_2\dot{\mathbf{d}}(t)dt \right\} \\
&\text{subject to} && \dot{\mathbf{z}}(t) = \mathbf{A}(\mathbf{z}(t))\mathbf{z}(t) + \mathbf{B}(\mathbf{p}(t) + \mathbf{u}(\mathbf{\Sigma A}, \mathbf{z})) \\
&&& \mathbf{z}(0) = \mathbf{0} \\
&&& \mathbf{\Sigma A}^{max} - \mathbf{\Sigma A} \geq \mathbf{0} \\
&&& \mathbf{\Sigma A} \geq \mathbf{0}
\end{aligned}
\tag{20}
$$

Since dealing with BRBs, $\mathbf{\Sigma A}$ stands for the total cross-sectional areas, which are limited by 0 and $\mathbf{\Sigma A}^{max}$. The two objective function forms in Equation (20) imply that the weighting matrix $\mathbf{Q}$ transforms $\mathbf{z}(t)$ into $\mathbf{d}(t)$ and $\dot{\mathbf{d}}(t)$. That is:

$$
\mathbf{Q} = \begin{bmatrix} (\mathbf{T}_{x \to d})' \mathbf{Q}_1 \mathbf{T}_{x \to d} & & & \\ & \mathbf{0} & & \\ & & (\mathbf{T}_{x \to d})' \mathbf{Q}_2 \mathbf{T}_{x \to d} & \\ & & & \mathbf{0} \end{bmatrix}
\tag{21}
$$

where $\mathbf{Q}_1$ and $\mathbf{Q}_2$ are diagonal weighting matrices whose components govern the minimization priority between $\mathbf{d}(t)$ and $\dot{\mathbf{d}}(t)$, respectively. The hysteretic components of $\mathbf{z}$ do not participate in the objective function.

The fixed-point iteration stems from the Lagrange function expression added by the initial condition $\mathbf{z}(0) = \mathbf{0}$ and the Karush–Kuhn–Tucker (KKT) conditions for $\mathbf{\Sigma A} - \mathbf{\Sigma A}^{max} \leq \mathbf{0}$ and $\mathbf{\Sigma A} \geq \mathbf{0}$. That is:

$$
\begin{aligned}
\mathcal{L}(\mathbf{z}, \mathbf{\Sigma A}, \lambda, \sigma, \mu) = \int_0^{t_f} H(t, \mathbf{z}(t), \mathbf{\Sigma A}, \lambda(t)) - \lambda'(t)\dot{\mathbf{z}}(t)dt + \dots \\
\mu_1'(\mathbf{\Sigma A} - \mathbf{A}^{max}) + \mu_2'(-\mathbf{\Sigma A}) + \sigma'(\mathbf{z}(0) - \mathbf{0})
\end{aligned}
\tag{22}
$$

where $\lambda(t)$ is the *4N*-dimensional time-varying Lagrange multipliers vector, $\sigma$ is a *4N*-dimensional multipliers vector that governs, $\mathbf{z}(0) = \mathbf{0}$, $\mu_1$ and $\mu_2$ are the KKT multipliers vector governing the design limitation inequality constraints, and H is the Hamilton function, which is given by:

$$
H(t, \mathbf{z}(t), \mathbf{\Sigma A}, \lambda(t)) = \mathbf{z}(t)'\mathbf{Q}\mathbf{z}(t) + \lambda'(t)[\mathbf{A}(t)\mathbf{z}(t) + \mathbf{B}(\mathbf{p}(t) + \mathbf{u}(t))]
\tag{23}
$$

Appendix A elaborates on the Lagrange function and develops the conditions for optimality. It ends with the following set of requirements:

Adjoint conditions :
$$\dot{\boldsymbol{\lambda}}(t) = -\nabla_{\mathbf{z}(t)}H(t, \mathbf{z}(t), \boldsymbol{\Sigma}\mathbf{A}, \boldsymbol{\lambda}(t))$$
$$\dot{\boldsymbol{\eta}}(t) = -\nabla_{\boldsymbol{\Sigma}\mathbf{A}}H(t, \mathbf{z}(t), \boldsymbol{\Sigma}\mathbf{A}, \boldsymbol{\lambda}(t))$$
Transversality conditions :
$$\boldsymbol{\lambda}(t_f) = \mathbf{0}$$
$$\boldsymbol{\eta}(t_f) = \mathbf{0}$$
Stationary condition :
$$\boldsymbol{\mu}_1 - \boldsymbol{\mu}_2 + \boldsymbol{\eta}(0) = 0$$
Complementarity condition :
$$\boldsymbol{\mu} = \left[ \begin{array}{c} \boldsymbol{\mu}_1 \\ \boldsymbol{\mu}_2 \end{array} \right] \geq \mathbf{0}, \ \boldsymbol{\mu}_1{}'(\boldsymbol{\Sigma}\mathbf{A} - \boldsymbol{\Sigma}\mathbf{A}^{max}) - \boldsymbol{\mu}_2{}'\boldsymbol{\Sigma}\mathbf{A} = 0, \quad \left[ \begin{array}{c} \boldsymbol{\Sigma}\mathbf{A} - \mathbf{A}^{max} \\ -\boldsymbol{\Sigma}\mathbf{A} \end{array} \right] \leq \mathbf{0}$$

$$(24)$$

where the function $\boldsymbol{\eta}(t)$ is defined as:

$$\boldsymbol{\eta}(t) = \int_t^{t_f} \nabla_{\boldsymbol{\Sigma}\mathbf{A}}H(t)d\tau \tag{25}$$

and the gradients of H in $\mathbf{z}(t)$ and $\boldsymbol{\Sigma}\mathbf{A}$ are:

$$\nabla_{\mathbf{z}(t)}H(t, \mathbf{z}(t), \boldsymbol{\Sigma}\mathbf{A}, \boldsymbol{\lambda}(t)) = 2\mathbf{Q}\,\mathbf{z}(t) + \left(\mathbf{A}(\mathbf{z}(t)) + \nabla_{\mathbf{z}(t)}\mathbf{A}(\mathbf{z}(t))'\mathbf{z}'(t) + \mathbf{u}'_{\mathbf{z}(t)}(\boldsymbol{\Sigma}\mathbf{A}, \dot{\mathbf{u}}, \mathbf{z}(t))\mathbf{B}'\right)\boldsymbol{\lambda}(t) \tag{26}$$

$$\nabla_{\boldsymbol{\Sigma}\mathbf{A}}H(t, \mathbf{z}(t), \boldsymbol{\Sigma}\mathbf{A}, \boldsymbol{\lambda}(t)) = \mathbf{u}'_{\boldsymbol{\Sigma}\mathbf{A}}(\boldsymbol{\Sigma}\mathbf{A}, \dot{\mathbf{u}}, \mathbf{z}(t))\mathbf{B}'\boldsymbol{\lambda}(t) \tag{27}$$

Regarding the gradient expressions, the control force derivatives $\mathbf{u}_{\mathbf{z}(t)}(\boldsymbol{\Sigma}\mathbf{A}, \dot{\mathbf{u}}, \mathbf{z}(t))$ and $\mathbf{u}_{\boldsymbol{\Sigma}\mathbf{A}}(\boldsymbol{\Sigma}\mathbf{A}, \dot{\mathbf{u}}, \mathbf{z}(t))$ are written as:

$$\mathbf{u}_{\mathbf{z}(t)}(\boldsymbol{\Sigma}\mathbf{A}, \dot{\mathbf{u}}, \mathbf{z}(t)) = -\mathbf{k}^{B,el}(\boldsymbol{\Sigma}\mathbf{A}) - \int_0^t \mathbf{k}^{B,hys}{}_{\mathbf{z}(\tau)}(\boldsymbol{\Sigma}\mathbf{A}, \mathbf{u}, \mathbf{z}(t))\mathbf{z}(\tau) + \mathbf{k}^{u,hys}(\boldsymbol{\Sigma}\mathbf{A}, \mathbf{u}, \mathbf{z})d\tau \tag{28}$$

$$\mathbf{u}_{\boldsymbol{\Sigma}\mathbf{A}}(\boldsymbol{\Sigma}\mathbf{A}, \dot{\mathbf{u}}, \mathbf{z}(t)) = -\mathbf{k}^{B,el}{}_{\boldsymbol{\Sigma}\mathbf{A}}(\boldsymbol{\Sigma}\mathbf{A})\mathbf{z}(t) - \int_0^t \mathbf{k}^{B,hys}{}_{\boldsymbol{\Sigma}\mathbf{A}}(\boldsymbol{\Sigma}\mathbf{A}, \mathbf{u}, \mathbf{z}(t))\mathbf{z}(\tau)d\tau$$
and :
$$\mathbf{k}^{B,el}{}_{\boldsymbol{\Sigma}\mathbf{A}}(\boldsymbol{\Sigma}\mathbf{A}) = (\mathbf{T}_{x\to\Delta})' \mathrm{diag}\left\{\left[\left(\mathbf{a}^B \odot \mathbf{E}^B \oslash \mathbf{L}^B\right)\mathbf{1}^S\right]\right\}\mathbf{T}_{x\to\Delta}$$
$$\mathbf{k}^{B,hys}{}_{\boldsymbol{\Sigma}\mathbf{A}}(\boldsymbol{\Sigma}\mathbf{A}, \mathbf{z}(t)) = (\mathbf{T}_{x\to\Delta})' \mathrm{diag}\left\{\left[(1 - \mathbf{a}^B) \odot \left(\rho^B(\sigma^B, \mathbf{x}, \dot{\mathbf{x}}) \odot \mathbf{E}^B \oslash \mathbf{L}^B\right)\mathbf{1}^S\right]\right\}\mathbf{T}_{x\to\Delta}$$

$$(29)$$

In Equation (28), the expression $\mathbf{G}_{\boldsymbol{\Sigma}\mathbf{A}}(\boldsymbol{\Sigma}\mathbf{A}, \mathbf{z}(\tau))$ is determined numerically.

The fixed-point iteration implementation is in discrete time. Accordingly, the satisfaction of the adjoint and transversality conditions is by solving $\dot{\mathbf{z}}(t) = \mathbf{A}(\mathbf{z}(t))\mathbf{z}(t) + \mathbf{B}(\mathbf{p}(t) + \mathbf{u}(\boldsymbol{\Sigma}\mathbf{A}, \dot{\mathbf{u}}, \mathbf{z}(t)))$, with the initial value $\mathbf{z}(0) = \mathbf{0}$, followed by solving in backward-time $\dot{\boldsymbol{\lambda}}(t) = -\nabla_{\mathbf{z}(t)}H(t, \mathbf{z}(t), \boldsymbol{\Sigma}\mathbf{A}, \boldsymbol{\lambda}(t))$ and $\dot{\boldsymbol{\eta}}(t) = -\nabla_{\boldsymbol{\Sigma}\mathbf{A}}H(t, \mathbf{z}(t), \boldsymbol{\Sigma}\mathbf{A}, \boldsymbol{\lambda}(t))$ with $\boldsymbol{\lambda}(t_f) = \mathbf{0}$ and $\boldsymbol{\eta}(t_f) = \mathbf{0}$. The complementarity term of Equation (24) is satisfied by defining the components of $\boldsymbol{\mu}_1$ and $\boldsymbol{\mu}_2$ as:

$$\eta_{1,n} = \begin{cases} 0 & \leftrightarrow & \boldsymbol{\Sigma}\mathbf{A}_n \neq \boldsymbol{\Sigma}\mathbf{A}_n^{max} \\ -\eta_n(0) & \leftrightarrow & \boldsymbol{\Sigma}\mathbf{A}_n = \boldsymbol{\Sigma}\mathbf{A}_n^{max} \end{cases} \quad \forall \quad n = 1, \dots, N \tag{30}$$

$$\eta_{2,n} = \begin{cases} 0 & \leftrightarrow & \boldsymbol{\Sigma}\mathbf{A}_n \neq 0 \\ \eta_n(0) & \leftrightarrow & \boldsymbol{\Sigma}\mathbf{A}_n = 0 \end{cases} \quad \forall \quad n = 1, \dots, N \tag{31}$$

That leaves us with the stationary term.

The introduction of the fixed-point iteration modifies $\boldsymbol{\Sigma}\mathbf{A}$ and leads $\dot{\boldsymbol{\eta}}$ to result in $\boldsymbol{\eta}(0)$ that satisfies the stationary term—reaching an optimal point. Substituting Equations (30) and (31) into the stationarity condition of Equation (24) gives:

$$-\boldsymbol{\eta}(0)'(\boldsymbol{\Sigma}\mathbf{A} - \boldsymbol{\Sigma}\mathbf{A}^{max}) = \mathbf{0} \tag{32}$$

The form of Equation (32) is ready for the fixed-point iteration applications so that the consequent iterative term is:

$$\boldsymbol{\Sigma}\mathbf{A}_n^{k+1} = \Pi_{\boldsymbol{\Sigma}\mathbf{A}}\left(\boldsymbol{\Sigma}\mathbf{A}_n^k + \gamma \boldsymbol{\eta}_n^k(0)\right) \quad \forall \quad n = 1, \dots, N \tag{33}$$



where $\gamma$ is the convergence parameter, $\Sigma A_n^k$ denotes the nth component of $\Sigma A^k$ at the kth iteration, and $\eta_n^k(0)$ denotes the nth component of $\eta^k(0)$ that stems from $\Sigma A^k$. The fixed-point iteration application is by the following procedure consisting of four initial steps and four iterative steps.

(i)    Set the iteration index to k = 0 and choose an initial $\Sigma A^0$
(ii)   Define the maximum control gains boundary vector $\Sigma A^{max}$
(iii)  Decide the weighting sub-matrices $Q_1$ and $Q_2$
(iv)   Choose $\gamma$ and define the number of maximum iterations $k^{max}$
(v)    Calculate $z^k(t)$ by solving the initial value problem:

$$\dot{z}(t) = A(z(t))z(t) + B\left(p(t) + u\left(\Sigma A^k, \dot{u}, z(t)\right)\right) \quad ; \quad z(0) = 0$$

(vi)   Calculate $\lambda^k(t)$ and $\eta^k(t)$ backward in time by solving:

$$\dot{\lambda}^k(t) = -\nabla_{z(t)}H\left(t, z(t), \Sigma A^k, \lambda^k(t)\right) \quad ; \quad \lambda^k(t_f) = 0$$

$$\dot{\eta}^k(t) = -\nabla_{\Sigma A}H\left(t, z(t), \Sigma A^k, \lambda^k(t)\right) \quad ; \quad \eta^k(t_f) = 0$$

(vii)  Update the components of $\Sigma A^k$ using the fixed-point iteration method:

$$\Sigma A_n^{k+1} = \Pi_{\Sigma A}\left(\Sigma A_n^k + \gamma \eta_n^k(0)\right) \quad \forall \quad n = 1, \ldots, N$$

(viii) Check if $-\eta^k(0)'\left(\Sigma A^{k+1} - \Sigma A^{max}\right) = 0$ is satisfied or $k = k^{max}$. If yes, then finish. If not, increase k by 1 and go back to step (v).

The eight-step procedure is a straightforward approach to solving Equation (20). Notice that step (vi) addresses the adjoint and transversality conditions, while step (vii) addresses the complementarity and stationary terms. The procedure stops when the stationary condition is satisfied.

## 5. Optimization Examples

Two optimization examples examine the fixed-point iteration application for inelastic rigid frame systems in a weakened state following an earthquake incident. The first example optimizes a low-rise three-story structure, and the second deals with a high-rise 15-story structure. The employed BRBs are IPN profiles of modulus of elasticity $E_{n,s}^B = 205,000$ MPa, axial yield stress of $\sigma_{n,s}^{B,Y} = 225$ MPa, and $a_{n,s}^B = 0.1$ ratios between the plastic and the elastic stiffness.

The numerical evaluation of the dynamic response is conducted using the Newmark-$\beta$ of the average acceleration scheme. The Newton–Raphson method is implemented in addressing the implicit hysteretic resisting and control forces. Equation (24) adjoint terms are calculated backward in discrete time based on the extended-mean-value theorem employed by Newmark-$\beta$.

### 5.1. Example 1: Three-Story Rigid Frame System

The three-story and two-spans rigid frame system's elevation scheme is depicted in Figure 5. The entire story's height is 4.5 m, the columns' effective length is 3.5 m, and the span's length is 4.0 m. The story stiffness quantities $k_{n=1,\ldots,3}^F$ in Figure 5 are calculated for clumped columns with squared cross-sections of 0.6 m $\times$ 0.6 m, 0.5 m $\times$ 0.5 m, and 0.4 m $\times$ 0.4 m dimensions in stories 1 to 3, respectively. The frame members' modulus of elasticity is set as 27,000 MPa, and the story masses are $m_{n=1,\ldots,3} = 40$ ton. The stories' yield force $f_n^{F,yld}$ specified in Figure 5 regard interstory drift of 0.5% of the columns' effective length, and the ratio between the plastic and the elastic stiffness is $a_n^F = 0.5$. The inherent damping ratio is assumed to be 5%, and the inherent damping matrix is calculated by Caughey damping.

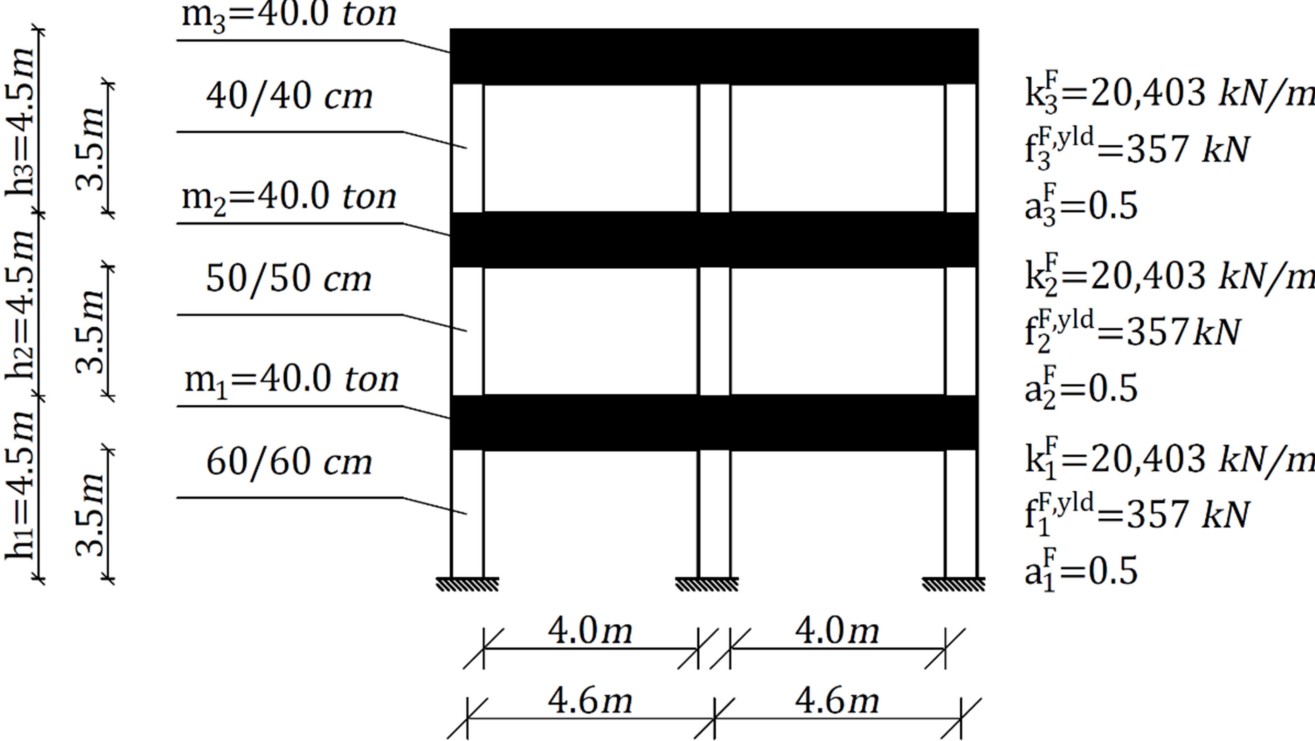

**Figure 5.** The bare three-story rigid frame system.

　　The highest modal period of the frame structure (without BRBs) is 0.87 s, a quantity that is considerably high for a three-story system—implying a weakened system. The initial BRBs allocation to start the fixed-point iteration consists of uniformly distributed IPN180 elements of cross-sectional area $A_{n,s}^0 = 27.9$ cm$^2$ and effective length $L_{n,s}^B \cong 5.3$ m. After the initial BRBs allocation, the highest modal period is reduced to $T_1 \cong 0.25$ s. This example's quasi-static cyclic loading is defined accordingly and is depicted in Figure 6.

　　The total cross-sectional areas corresponding to the IPN180 profiles and the static gains to be optimized are $\Sigma A_1^0 = \Sigma A_2^0 = \Sigma A_3^0 \cong 55.8$ cm$^2$s. The static gains maximum corresponds to the IPN600 profile of cross-sectional area 254 cm$^2$, hence $\Sigma A_1^{max} = \Sigma A_2^{max} = \Sigma A_3^{max} = 508$ cm$^2$. The weighting matrices are defined by default as $\mathbf{Q}_1 = \mathbf{I}$ and $\mathbf{Q}_2 = \mathbf{I}$. The iterative convergence parameters are set for each story individually so that of $\gamma_1 = 0.007$, $\gamma_2 = 0.005$, $\gamma_3 = 0.002$, and $k^{max} = 50$. That concludes the algorithm's preparation steps (i)–(iv).

　　Following the preparation steps, the iterative process is initiated in steps (v)–(viii). Figure 7 shows the iterative process of $\Sigma A_n^k$ versus the iteration index $k = 0, 1, \ldots, 50$, which converged to static gains of $\Sigma A_1^{50} \cong 177.0$ cm$^2$, $\Sigma A_2^{50} \cong 191.0$ cm$^2$, and $\Sigma A_1^{50} \cong 161.0$ cm$^2$. The IPN allocation corresponding to the optimal static gains consists of IPN340 in the first story (i.e., $A_{s=1,2,n=1} = 86.7$ cm$^2$), IPN360 in the second story (i.e., $A_{s=1,2,n=2} = 97.0$ cm$^2$), and IPN320 in the third story (i.e., $A_{s=1,2,n=3} = 77.7$ cm$^2$). The elevation scheme of the optimal BRBs allocation is depicted in Figure 8b.

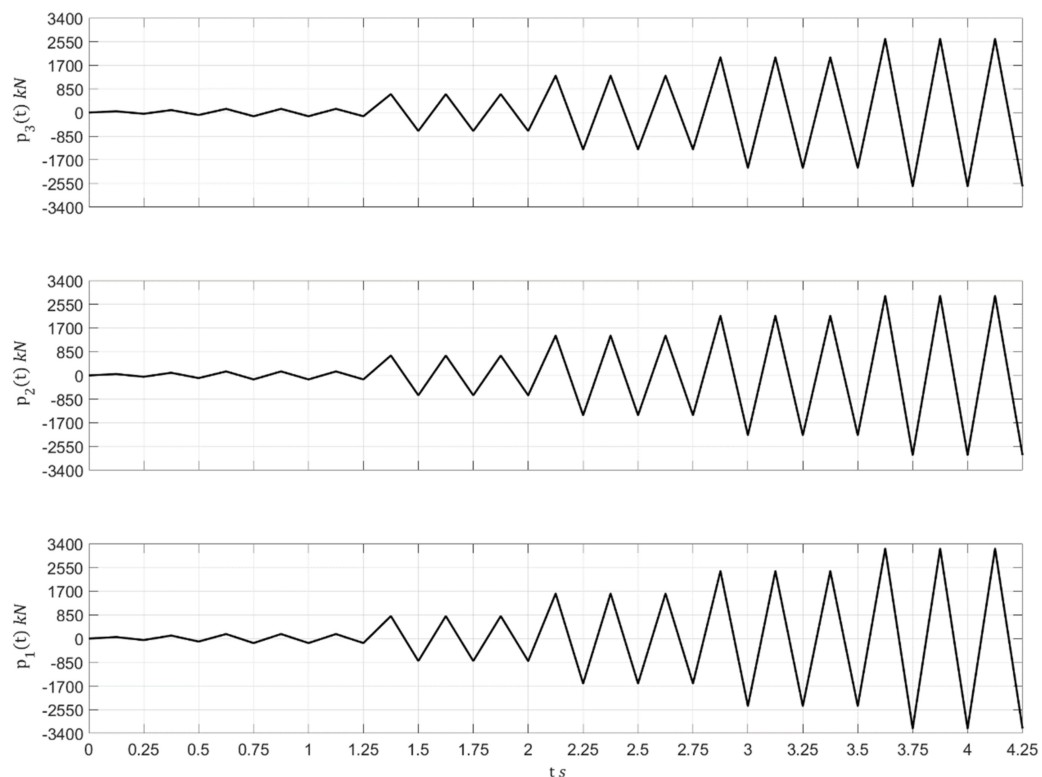

**Figure 6.** Example 1 quasi-static cyclic loading.

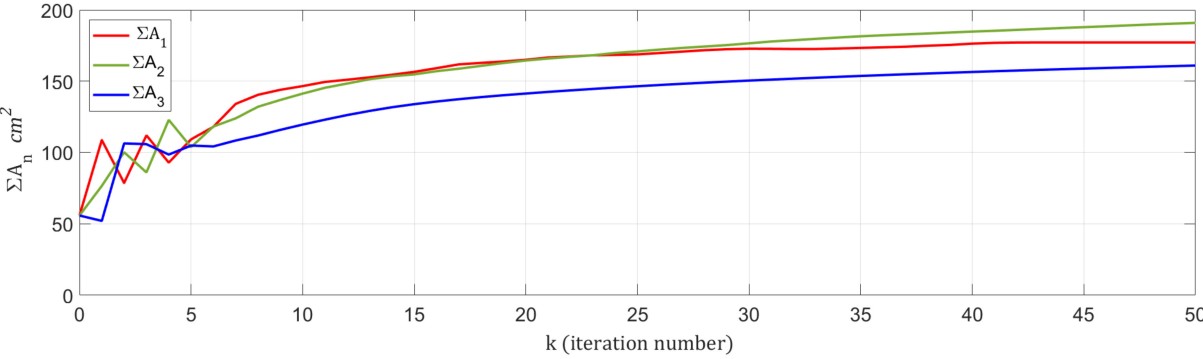

**Figure 7.** Fixed-point iterative process for the three-story rigid frame system.

The optimal static gains quantity is also examined for equivalent uniform IPN profile distribution. The equivalency is in terms of total cross-sectional area. In a case where modifying the fixed-point iteration's distribution with uniform distribution increases the objective function, it indicates that the solution is indeed optimal. Accordingly, the members' cross-sectional area is calculated by:

$$\Sigma A_1^{50} + \Sigma A_2^{50} + \Sigma A_1^{50} \cong 529.0 \text{ cm}^2 \quad \rightarrow \quad A_{s,n} = \frac{529}{6} \cong 88.0 \text{ cm}^2 \quad \forall \quad n = 1, \ldots, 3 \quad \& \quad s = 1, 2$$

The quantity $A_{s,n} = 88.0 \text{ cm}^2$ correlates to the IPN340 profile and corresponds to the uniform BRBs distribution depicted in Figure 8c.

Figure 9 illustrates the portions of Equation (20) objective function $\int_0^{t_f} \mathbf{d}^T(t) d(t) dt$ and $\int_0^{t_f} \dot{\mathbf{d}}^T(t) \dot{\mathbf{d}}(t) dt$, in t, given that $\mathbf{Q}_1 = \mathbf{I}$ and $\mathbf{Q}_2 = \mathbf{I}$. Figure 8 depicts the initial IPN distribution, the optimal IPN distribution, and the equivalent uniform IPN distribution under Figure 6 loading. The plot exemplifies that the objective function is significantly reduced and that changing the optimal distribution with a uniform distribution across all stories slightly increases the objective function, which suggests the fixed-point iteration design is indeed optimal.

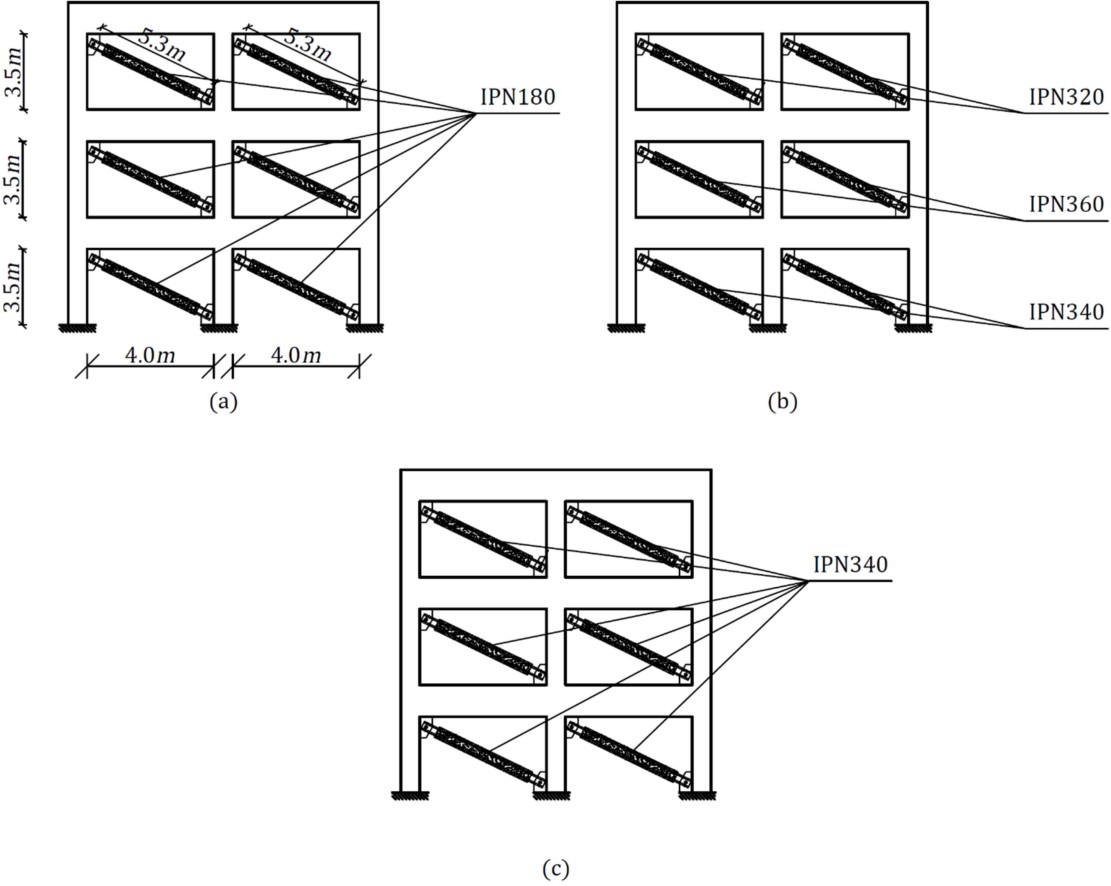

**Figure 8.** Three-story inelastic rigid frame system configurations: (**a**) initial BRBs allocation, (**b**) optimal BRBs allocation, and (**c**) equivalent uniform BRBs distribution.

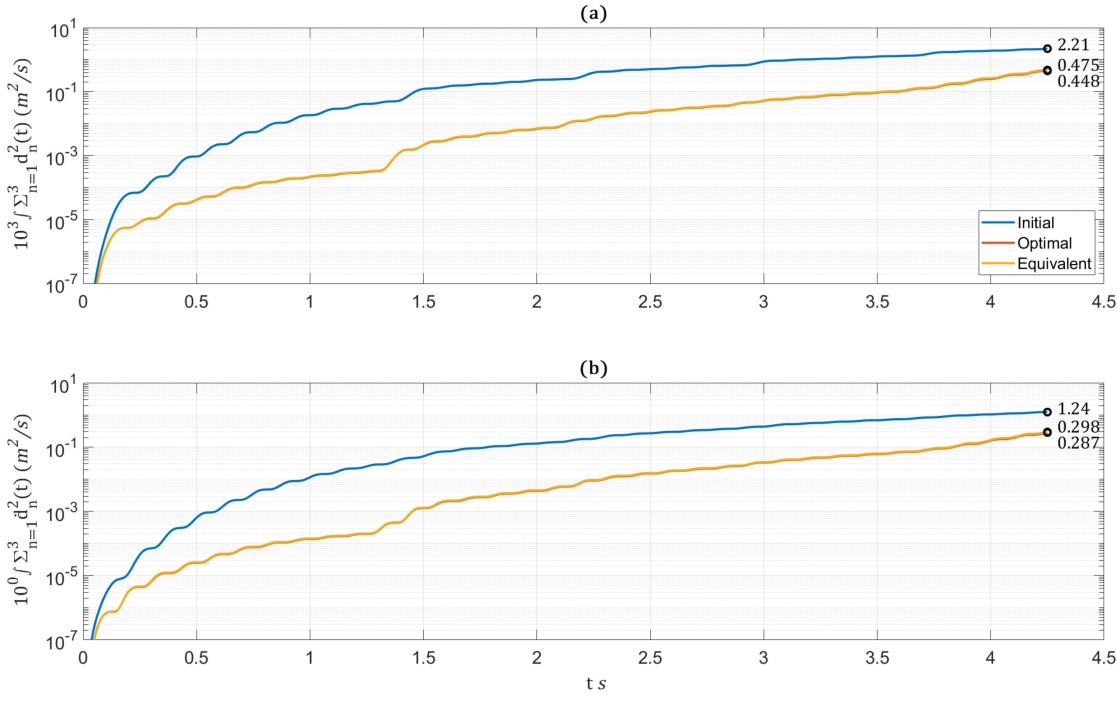

**Figure 9.** Trajectories of the three-story inelastic rigid frame system: (**a**) interstory drifts (**b**) interstory drift velocities.

The earthquake response of the three IPN allocation possibilities is examined under the Valparaiso 2017 ground acceleration—recorded by the Torpederas station (east–west component) of 0.91 g's peak ground acceleration. Figure 10 depicts the stories' hysteretic resisting forces versus the relative interstory drift. The employment of BRBs maintained the columns by remaining in their elastic regime while the BRBs slightly met the plastic range. These substantial results are thanks to addressing the quasi-static cyclic loading of twice the total yielding force maximum amplitude.

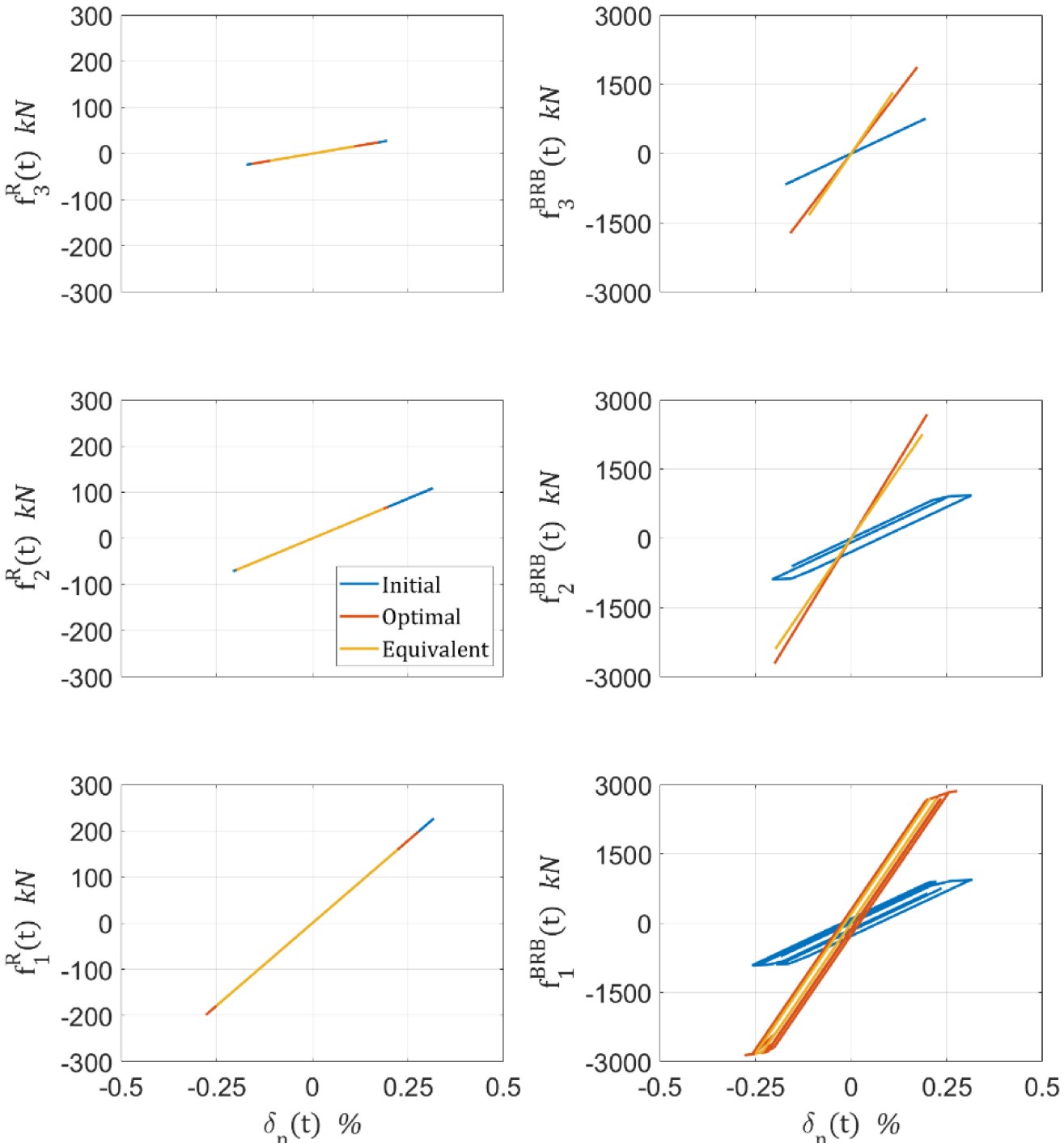

**Figure 10.** Hysteretic resisting forces of the three-story frame system under the Valparaiso 2017 earthquake.

### 5.2. Example 2: 15-Story Rigid Frame System

The second example deals with regulating the dynamic response of the 75.0 m high 15-story frame system depicted in Figure 11. The stories' height is 5.0 m, and the spans' length is 7.0 m. The clumped columns are of 4.0 m effective length, and their variation in column cross-sectional dimensions is specified in Figure 11. The material properties of the BRBs and Columns are the same as in Example 1. The BRBs' effective length is $L_{n,s}^B \cong 8.6$—considering the lateral angle of approximately $45.0°$. The quantities of the ceilings' mass $m_n$, the lateral story stiffnesses $k_n^F$, the columns' shear force at first yield $f_n^{F,yld}$, and the ratio between the plastic and the elastic stiffness $a_n^F$ are specified in Figure 11 as well.

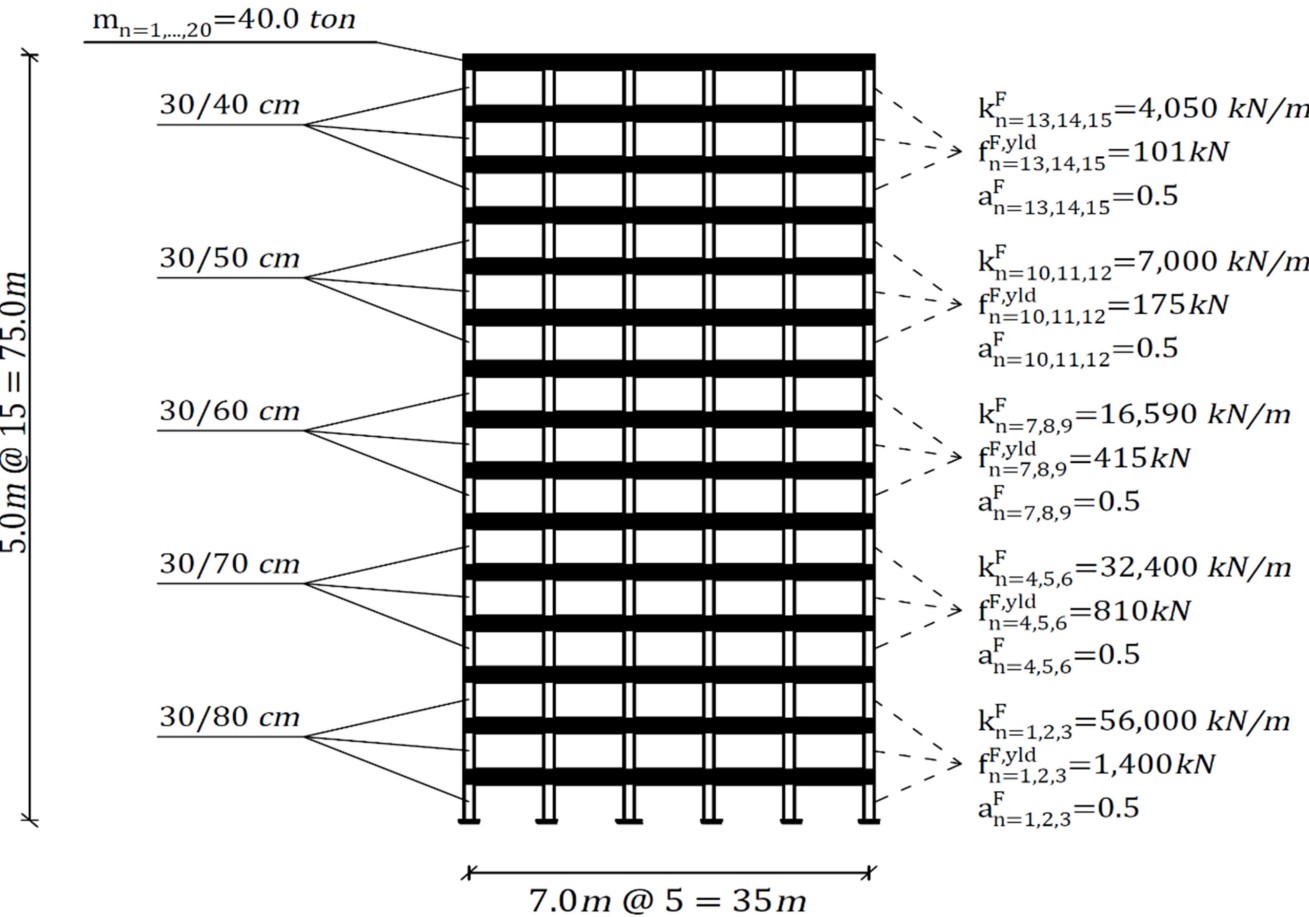

**Figure 11.** Fifteen-story inelastic rigid frame system.

The dominant undamped free-vibrations period is approximately $T_1 = 0.92$ s. In this example, at the initial iteration, IPN180 profiles are assigned to all spans and stories, which provides $\Sigma A_1^0 = \ldots = \Sigma A_{15}^0 \cong 139.5$ cm$^2$s. Considering the IPN600 profile as the maximum limitation on the gain variables, we have $\Sigma A_1^{max} = \ldots = \Sigma A_{15}^{max} = 1270$ cm$^2$. The initial frame system equipped with BRBs is depicted in Figure 12a, and the related quasi-static cyclic loadings are illustrated in Figure 13.

The algorithm convergence parameter is decided by $\gamma = 10^{-2}$. The fixed-point iteration method goes under the iterative process shown in Figure 14. At the $k = 50$ iteration, the algorithm converged into the optimal solution whose resultant optimal gain variables (i.e., total cross-sectional IPN areas) correspond to the BRBs allocation scheme depicted in Figure 12b. Additionally, the total sum of optimal gains is calculated and divided uniformly to all frame spans, yielding the distribution in Figure 12c to showcase that Figure 12b is indeed optimal.

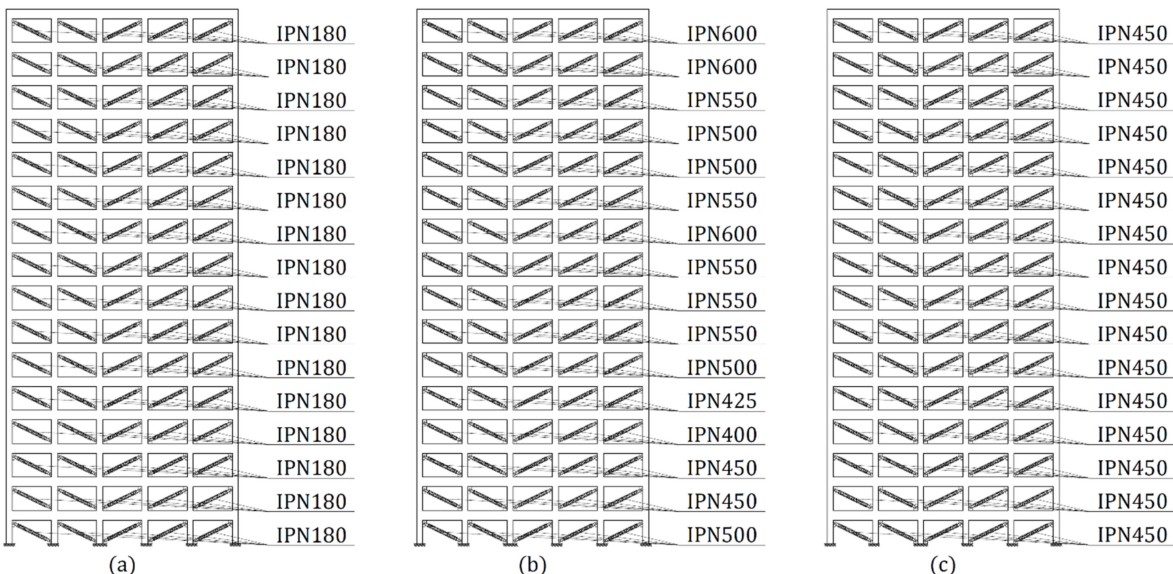

**Figure 12.** Fifteen-story rigid frame system configurations: (**a**) initial BRBs allocation (**b**) optimal BRBs allocation (**c**) equivalent uniform BRBs distribution.

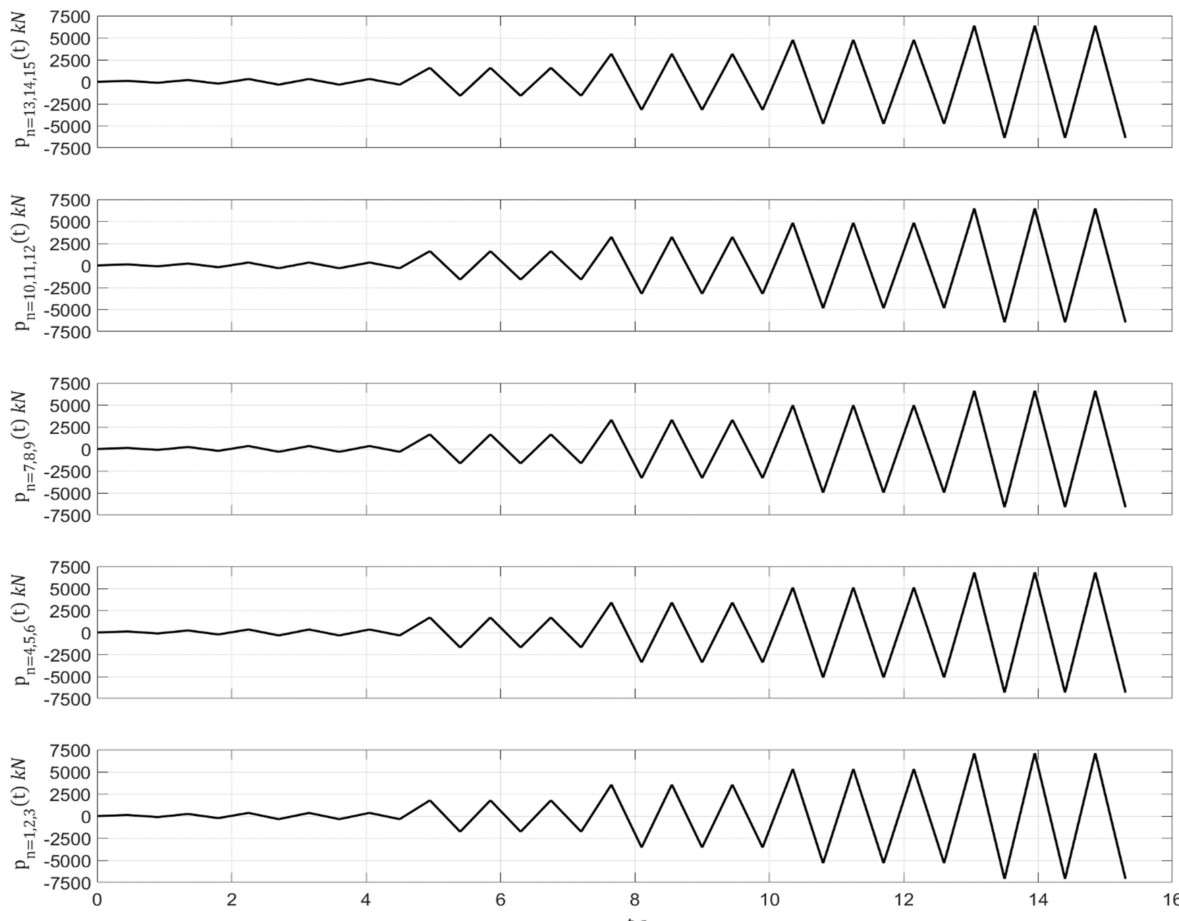

**Figure 13.** Example 2 quasi-static cyclic loadings.

Figure 15 shows the trajectories of $\int_0^{t_f} \mathbf{d}^T(t)\mathbf{d}(t)dt$ and $\int_0^{t_f} \dot{\mathbf{d}}^T(t)\dot{\mathbf{d}}(t)dt$, as portions of the objective function, for the initial, optimal, and equivalent uniform BRB distributions. It is shown that the optimal distribution provides the minimum objective function—indicating the solution integrity. The three distributions are also examined for the Valparaiso 2017

earthquake of 0.91 g's PGA. Figure 16 shows the hysteretic behavior of stories 1,4,7,10, and 13 regarding the columns' and BRBs' hysteretic forces. The forces exemplify that the frame system remains in its elastic range regardless of the strong earthquake.

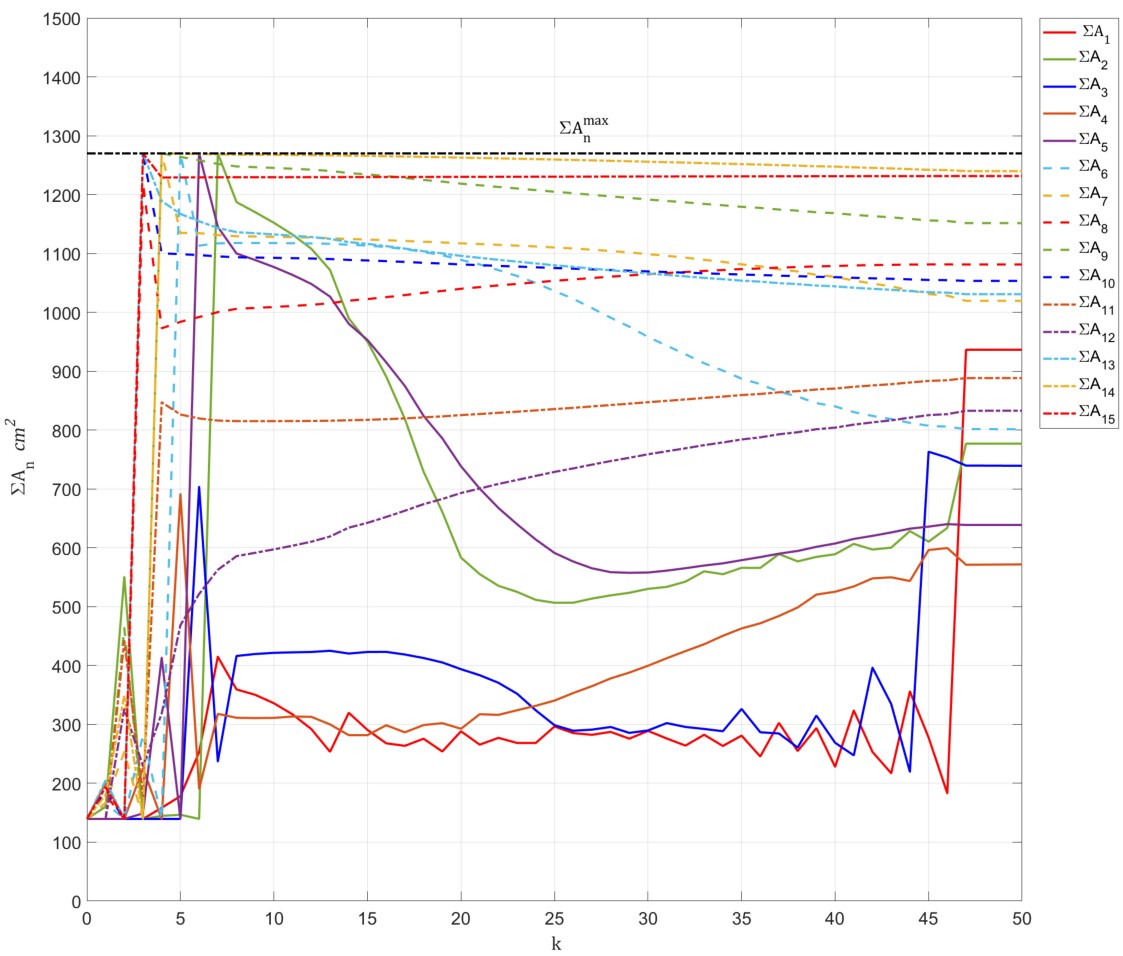

**Figure 14.** Fixed-point iterative process for the 15-story rigid frame system.

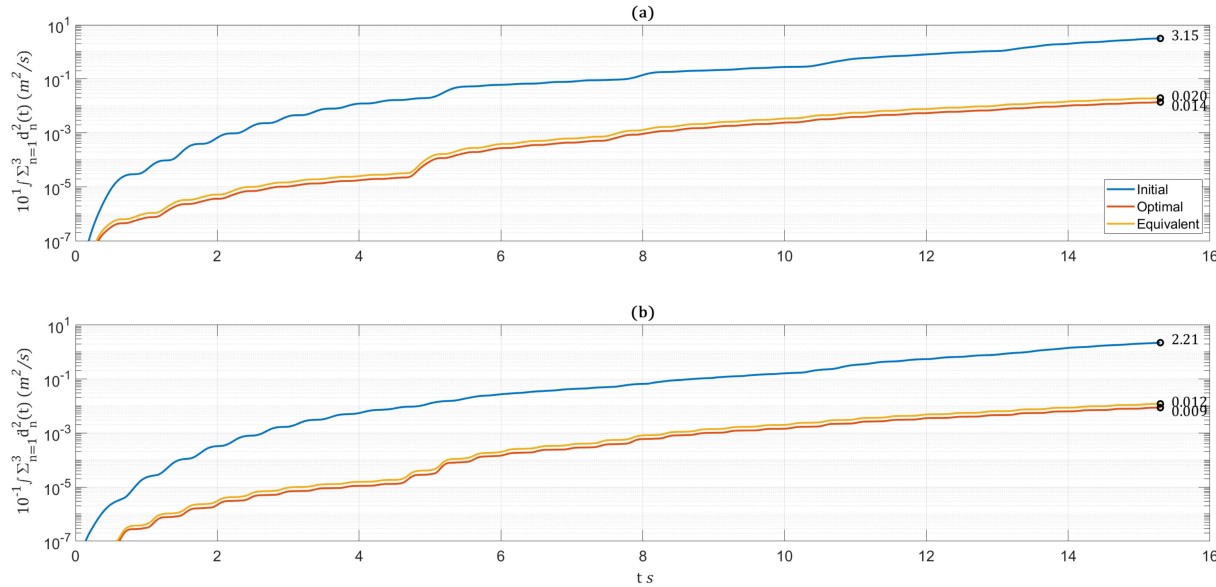

**Figure 15.** Trajectories of the 15-story inelastic rigid frame system: (**a**) interstory drift (**b**) interstory drift velocities.

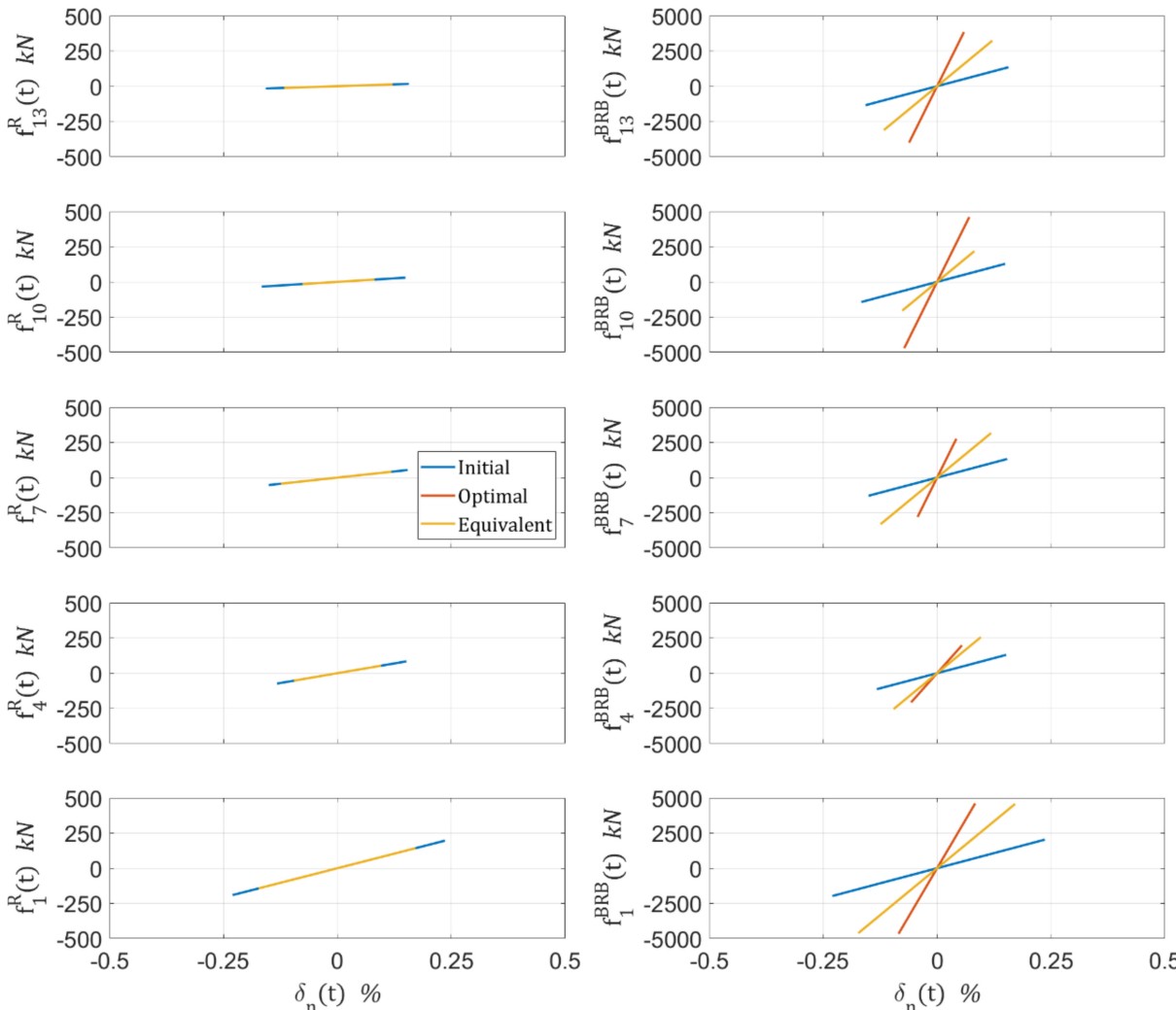

**Figure 16.** Hysteretic resisting forces of the 15-story frame system under the Valparaiso 2017 earthquake.

## 6. Conclusions

The paper presents a practical optimization procedure for retrofitting frame structure post-earthquake event. The optimization addresses nonlinear control systems whose mechanism consists of linear, nonlinear, and hysteretic portions. Such control systems are characterized by static parameters (static gains)—referring to the control system's geometric or material characteristics. This paper's optimization procedure is the first to utilize the fixed-point iteration method for controlling and regulating the dynamic response of frame structures.

The state-space equation, associated with the fixed-point iteration, defines the dynamic equilibrium. It is derived from the frame structure's lateral force equilibrium and regards a closed-loop paradigm with negative state feedback and the controller—consisting of linear, nonlinear, and hysteretic portions. The nonlinear control system minimizes the cumulation sum of squared interstory drifts deformations and velocities by calculating the optimal gains while subject to design boundaries. The solution procedure comprises four initial and four iterative steps that are repeated until all optimality conditions are satisfied or the maximum number of iterations has been reached.

The fixed-point iteration scheme presented in this paper differs from other control algorithms in being suitable to address linear, nonlinear, and hysteretic control law. The BRB system is employed in showcasing the application of the developed methodology. Choosing the BRB system is due to having linear and hysteretic portions. Two optimization

examples address weakened frame systems (following earthquake incidents) and demonstrate the design procedure practicability and illustrate the fixed-point iterative capability in optimizing multiple control gain variables. It should be noted that the fixed-point iteration converges into local optima. Thus, each of the first four steps of the solution procedure (i.e., deciding on the weighting matrices, choosing the convergence parameter, defining the maximum control gains, and setting the initial control gain variables) significantly influences the final solution.

In closing, the methodology of this paper help to optimize the static specifications of control systems that produce either linear, nonlinear, or hysteretic forces to regulate the seismic vibrations of inelastic systems. The static specifications relate to geometrical, strength, or material parameters. While this paper addresses weakened frame structures, the methodology is relevant to any lateral-load resisting force system whose state vector is calculated.

**Author Contributions:** Conceptualization, A.S. and M.G.; methodology, A.S.; software, A.S.; validation, A.S.; formal analysis, A.S.; investigation, A.S.; resources, A.S.; data curation, A.S.; writing—original draft preparation, A.S. and M.G; writing—review and editing, A.S.; visualization, A.S.; supervision, A.S.; project administration, M.G.; funding acquisition, M.G. All authors have read and agreed to the published version of the manuscript.

**Funding:** This research received no external funding.

**Data Availability Statement:** The data presented in this study are available on request from the corresponding author.

**Conflicts of Interest:** The authors declare no conflict of interest.

## Abbreviations

| | |
|---|---|
| $\mathbf{a}^{\mathrm{B}}$ | $N \times S$ matrix consisting of the ratios between the BRBs' axial plastic and elastic stiffnesses |
| $\mathbf{c}$ | $N \times N$ inherent damping matrix |
| $\mathbf{c}^{\mathrm{u,el}}$ | $N \times N$ linear damping matrix of $\mathbf{u}$ |
| $\mathbf{d}$ | $N$-dimensional interstory drifts vector |
| $\mathbf{f}^{\mathrm{u,NL}}$ | $N$-dimensional nonlinear portion of $\mathbf{u}$ |
| $\mathbf{f}^{\mathrm{F}}$ | $N$-dimensional structural rigid frame system's lateral resisting force vector |
| $\mathbf{f}^{\mathrm{F,hys}}$ | $N$-dimensional hysteretic portion of $\mathbf{f}^{\mathrm{F}}$ |
| $f_{\mathrm{n}}^{\mathrm{F,yld}}$ | nth story columns' shear force at first yield |
| k | Fixed-point iteration number |
| $\mathrm{k}^{\mathrm{max}}$ | Fixed-point maximum iteration number |
| $\mathbf{k}^{\mathrm{u,el}}$ | $N \times N$ linear stiffness matrix of $\mathbf{u}$ |
| $\mathbf{k}^{\mathrm{u,hys}}$ | $N \times N$ hysteretic stiffness matrix of $\mathbf{u}$ |
| $\mathbf{k}^{\mathrm{F,el}}$ | $N \times N$ static-condensate matrix elastic stiffness portion about $\mathbf{d}$ |
| $\mathbf{k}^{\mathrm{F,hys}}$ | $N \times N$ static-condensate hysteretic stiffness matrix |
| $\mathbf{m}$ | $N \times N$ static-condensate diagonal mass matrix |
| $\mathbf{p}$ | $N$-dimensional lateral applied dynamic load vector |
| $\mathbf{u}$ | $N$-dimensional control force vector |
| $\mathbf{x}$ | $N$-dimensional ceilings' relative-to-ground displacement vector |
| $\dot{\mathbf{x}}$ | $N$-dimensional ceilings' relative-to-ground velocities vector |
| $\ddot{\mathbf{x}}$ | $N$-dimensional ceilings' relative-to-ground accelerations vector |
| $\mathbf{z}(\mathrm{t})$ | $4N$-dimensional state-vector |
| $\mathbf{A}$ | $4N \times 4N$ state matrix |
| $\mathbf{A}^{\mathrm{B}}$ | $N \times S$ matrix containing the BRBs' effective cross-sectional area |
| $A_{n,s}^{B}$ | cross-sectional area of the BRB installed at the sth span of the nth story |
| $\mathbf{B}$ | $4N \times N$ input-to-state matrix |
| $\mathbf{E}^{\mathrm{B}}$ | $N \times S$ matrix composed of the BRBs' elasticity modulus |
| $\mathbf{G}$ | $N \times 4N$ gain matrix |
| $H$ | Hamilton function |

| $\mathbf{L}^B$ | $N \times S$ matrix comprised of the BRBs' length |
|---|---|
| $\mathbf{Q}_1$ | diagonal weighting matrix whose components govern the minimization priority of $\mathbf{d}(t)$ |
| $\mathbf{Q}_2$ | diagonal weighting matrix whose components govern the minimization priority of $\dot{\mathbf{d}}(t)$ |
| $\mathbf{T}_{d \to x}$ | transformation matrix from drift coordinates into displacement coordinates |
| $\mathbf{T}_{x \to d}$ | transformation matrix from displacement coordinates into drift coordinates |
| $\mathbf{T}_{x \to \Delta}$ | geometric transformation matrix from $\mathbf{x}$ into the BRB's axial deformation $\mathbf{\Delta}$ |
| $T_1$ | highest modal period |
| $\mathbf{T}^{hys}$ | transformation matrix applied to the negative feedback of $\mathbf{z}$ to yield the hysteretic portion of $\mathbf{u}$ |
| $\mathbf{T}^{lin}$ | transformation matrix applied to the negative feedback of $\mathbf{z}$ to yield the linea portion of $\mathbf{u}$ |
| $\mathbf{T}^{NL}$ | transformation matrix applied to the negative feedback of $\mathbf{z}$ to yield the nonlinear portion of $\mathbf{u}$ |
| $\mathbf{\lambda}$ | $4N$-dimensional Lagrange multipliers vector |
| $\mathbf{\mu}_1$ | KKT multipliers vector governing the design limitation inequality constraints |
| $\mathbf{\mu}_2$ | KKT multipliers vector governing the design limitation inequality constraints |
| $\mathbf{\rho}^B$ | $N \times S$ matrix that corresponds to the material's elastic/plastic/unloading stages |
| $\mathbf{\sigma}$ | $4N$-dimensional Lagrange multipliers vector that governs initial conditions |
| $\mathbf{\sigma}^{in}$ | $N \times S$ matrix of the inelastic portion of the BRBs' axial stress |
| $\mathbf{\sigma}^{B,Y}$ | $N \times S$ matrix of the BRBs' yield stress |
| $\dot{\mathbf{\Delta}}^B$ | $N \times S$ matrix of BRBs' axial deformation rate vector |
| $\mathbf{\Sigma A}$ | $N$-dimensional vector consisting of the static gain variables |
| $\mathbf{\Sigma A}^{max}$ | $N$-dimensional vector consisting of the maximum allowable static gains |
| $\mathcal{L}$ | Lagrange function |
| $\mathbf{0}$ | $N$-dimensional vector of zeros |
| $\mathbf{1}^S$ | $S$-dimensional vector of ones |

## Appendix A

1.  Given the optimization problem:

$$\underset{\mathbf{\Sigma A}}{\text{minimize}} \quad \left\{ J = \int_0^{t_f} \mathbf{z}^T(t)\mathbf{Q}\mathbf{z}(t)dt \right\}$$

2.  $$\text{subject to} \quad \begin{aligned} &\dot{\mathbf{z}}(t) = \mathbf{A}(\mathbf{z}(t))\mathbf{z}(t) + \mathbf{B}(\mathbf{p}^{max}\sin\omega_1 t + \mathbf{u}(\mathbf{\Sigma A}, \mathbf{z})) \\ &\mathbf{z}(0) = \mathbf{0} \\ &\mathbf{\Sigma A}^{max} - \mathbf{\Sigma A} \geq \mathbf{0} \\ &\mathbf{\Sigma A} \geq \mathbf{0} \end{aligned} \quad \text{(A1)}$$

3.  In reference to Theorem 2.3.24 in Chapter 2 in the book of Gerdts [44]. Assume $\mathbf{z}^*(t)$ is the optimal trajectory of the state vector and $\mathbf{\Sigma A}^*$ composed of the optimal control gains, then there exists $\mathbf{\lambda}(t)$, $\mathbf{\sigma}$, and $\mathbf{\mu}$ such that:

4.  $\nabla_{\mathbf{z},\mathbf{\Sigma A}}\mathcal{L}(\mathbf{z}^*, \mathbf{\Sigma A}^*, \mathbf{\lambda}, \mathbf{\sigma}, \mathbf{\mu}) = \mathcal{L}_{\mathbf{z}}(\mathbf{z}^*, \mathbf{\Sigma A}^*, \mathbf{\lambda}, \mathbf{\sigma}, \mathbf{\mu})\mathbf{\Delta z} + \mathcal{L}_{\mathbf{\Sigma A}}(\mathbf{z}^*, \mathbf{\Sigma A}^*, \mathbf{\lambda}, \mathbf{\sigma}, \mathbf{\mu})\mathbf{\Delta\Sigma A} = 0$ (A2)

5.  Where $\mathbf{\Delta z}(t)$ and $\mathbf{\Delta\Sigma A}$ Denote small changes to the optimal solution of $\mathbf{z}^*(t)$ and $\mathbf{\Sigma A}^*$:

6.  $\mathbf{\Delta z}(t) = \mathbf{z}(t) - \mathbf{z}^*(t) \approx \mathbf{0}$ (A3)

7.  $\mathbf{\Delta\Sigma A} = \mathbf{\Sigma A} - \mathbf{\Sigma A}^* \approx \mathbf{0}$ (A4)

8.  Then, Equation (A2) implies that the small changes result in:

9.  $\mathcal{L}_{\mathbf{z}}\mathbf{\Delta z} = \mathbf{\sigma}' \mathbf{\Delta z}(0) + \int_0^{t_f}\left(\nabla_{\mathbf{z}(t)}\mathcal{H}(t)\right)\mathbf{\Delta z}(t)dt - \int_0^{t_f}\mathbf{\lambda}'(t)\mathbf{\Delta\dot{z}}(t)dt = 0$ (A5)

10. $\mathcal{L}_{\mathbf{\Sigma A}}\mathbf{\Delta\Sigma A} = \left(\mathbf{\mu}_1 - \mathbf{\mu}_2 + \int_0^{t_f}\nabla_{\mathbf{\Sigma A}}\mathcal{H}(t)dt\right)\mathbf{\Delta\Sigma A} = 0$ (A6)

11. Equation (A5) is further elaborated by using integration by parts to replace $\int_0^{t_f}\mathbf{\lambda}'(t)\mathbf{\Delta\dot{z}}(t)dt$ and deriving:

12. $\mathcal{L}_{\mathbf{z}}d\mathbf{z} = \mathbf{\lambda}(t_f) - (\mathbf{\lambda}(0) - \mathbf{\sigma})^T\mathbf{\Delta z}(0) + \int_0^{t_f}\left(\nabla_{\mathbf{z}}\mathcal{H}(t) + \dot{\mathbf{\lambda}}(t)\right)\mathbf{\Delta z}(t)dt = 0$ (A7)

13. Accordingly, the following adjoint conditions and transversality conditions have to be satisfied to comply with Equation (A7):

14. $\dot{\mathbf{\lambda}}(t) = -\nabla_{\mathbf{z}}\mathcal{H}(t, \mathbf{z}^*(t), \mathbf{\Sigma A}^*, \mathbf{\lambda}(t))$ (A8)

15. $\mathbf{\lambda}(t_f) = \mathbf{0}$ (A9)

16. Also, the definition for $\mathbf{\sigma}$ is:

17. $\mathbf{\sigma} = \mathbf{\lambda}(0)$ (A10)

18. Address the equality of Equation (A6) by defining the time function $\mathbf{\eta}(t)$:

19. $\mathbf{\eta}(t) = \int_t^{t_f}\nabla_{\mathbf{\Sigma A}}\mathcal{H}(\tau)d\tau$ (A11)

20. Thus, having additional adjoint and transversality conditions:

21. $\dot{\boldsymbol{\eta}}(t) = -\nabla_{\boldsymbol{\Sigma A}}\mathcal{H}(t, \mathbf{z}^*(t), \boldsymbol{\Sigma A}^*, \boldsymbol{\lambda}(t))$ (A12)

22. $\boldsymbol{\eta}(t_f) = 0$ (A13)

23. Consequently, Equation (A6) provides the stationary condition:

24. $\boldsymbol{\mu}_1 - \boldsymbol{\mu}_2 + \boldsymbol{\eta}(0) = 0$ (A14)

25. Thus, having the necessary conditions of Equation (24) regarding Equation (20) while requiring the KKT complementarity conditions:

26. $\boldsymbol{\mu} = \begin{bmatrix} \boldsymbol{\mu}_1 \\ \boldsymbol{\mu}_2 \end{bmatrix} \geq \mathbf{0} : \boldsymbol{\mu}_1^T(\boldsymbol{\Sigma A} - \boldsymbol{\Sigma A}^{max}) - \boldsymbol{\mu}_2^T(\boldsymbol{\Sigma A}) = \mathbf{0} \leftrightarrow \begin{bmatrix} \boldsymbol{\Sigma A} - \boldsymbol{\Sigma A}^{max} \\ -\boldsymbol{\Sigma A} \end{bmatrix} \leq \mathbf{0}$ (A15)

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
