# Peer review of "A Design Methodology for the Seismic Retrofitting of Existing Frame Structures Post-Earthquake Incident Using Nonlinear Control Systems"

_buildings, doi:10.3390/buildings12111886_

Round 1
Reviewer 1 Report
The research is a well-prepared and complex assessment of a very important topic. Determining the optimal BRB system for strengthening weakened structures in earthquakes a key role in safety.
Please reference the applied methods, also in part 2 and 4 of the paper. The equations are not new just presented. The novelty is to use the fixed-point iteration for this case. Also in case of equations, please make it clear the author´s contribution.
Figures need improvement. Try to make the size of letter similar: readeable (e.g. Fig.2 and similar figures) and not too big (Fig. 3, Fig. 5). Some figures are blured (e.g. Fig 4). Use dotted and dashed lines for Fig. 7 and similar figures (Fig. 9, Fig , and try to make it more clear when they are overlapong. Please label axes and give unit of measure everywhere. After FIg. 10 should come Fig. 11 not Fig 3. All figures need to be explained in text, especially the last one (obviuosly not Fig. 5, maybe Fig 14).
As a conclusion I would suggest to publish the paper with minor revisions (better quality figures, maybe the already used references also in part 2 and 4 of the papar, English language editing mostly in first part of the paper:abstract and introduction, and conclusion).
Reviewer 2 Report
Manuscript number: buildings-1947132
Title: A design methodology for the seismic retrofitting of existing frame
structures post-earthquake incident using nonlinear control systems
Comments of the reviewer:
Generally coments:
The proposed paper presents a new optimization algorithm for non-linear control systems, using fixed-point iteration to optimize their gains for optimal dynamic performance. However, the frame case with BRBs is inadequate and the algorithm only optimizes the cross-sectional area of BRBs. Thus, this article also needs to be supplemented with some additional work.
Specific comments:
(1) Why only the maximum fundamental vibrations period of the structure is used as a global damage index? In many research, the global damage indices may be inter-story drift angle, energy dissipation, or other parameters. The literature on global damage indices is still not sufficiently summarized.
(2) The quasi-static cyclic loading was used to assess the seismic performance of the frame structures with BRBs before and after optimization, but in engineering, it would be more straightforward and effective to use a dynamic time analysis.
(3) The innovation of the research still needs to be highlighted and how the proposed control algorithm differs from and has advantages over others.
(4) The structure improving technique could be viscous damper or tuned mass damper, so why choose only the structure with BRBs as a case study? Perhaps two and more structures could have been chosen as optimization examples.
(5) It is hard to clearly distinguish the relative-to-ground displacements xi and the shape of the undeformed structure indicated by the dashed line in Figure 1. It needs to be re-plotted.
(6) Equations (8) to (11) are perhaps missing acceleration terms and need to be checked. Besides, Eq. (19) shows Thys = Tlin, there is an error indicated here.
(7) There is a problem with the image reference on line 160 and a similar problem with the content that follows, check again for the article format and the graphic numbering.
(8) What do the connection symbols in Figure 3 mean? It is also necessary to add a corresponding explanation of the closed-loop control process.
(9) Line 303 states that Caughey damping is used, but the Rayleigh damping is the most widely used damping model, how was the damping model chosen?
(10) All diagrams should be optimized and the parameters in the text can be presented in table form.
(11) In line 398, the iteration curve fluctuates significantly, why does this happen, and does the calculation converge after 50 iterations?
(12) From the diagram shown on line 393, is it necessary to arrange BRBs at each span? The proposed algorithm can optimize the cross-sectional area of the BRBs, but can the number of BRBs be optimally calculated? And in line 402, it is clear from the diagram that the BRBs are not functioning and are still in an elastic state, indicating that too many BRBs are arranged.
Reviewer 3 Report
This manuscript presented a design methodology for the seismic retrofitting of existing frame structures. The manuscript needs thorough corrections as specified in the following comments:
1. There are many sentences in the text that have errors in grammar and should be corrected. The authors suggest doing a proof of English reading and editing a manuscript to correct all grammar errors.
2. The abstract has a poor presentation; it should be re-written to include the studied parameters, the methodology, and the important findings.
4. On what basis do the authors consider the range or level of each of the Example 1: Three-story rigid frame system and Example 2: 15-story rigid frame system.? Would you please elaborate on the manuscript to better understand the readers?
6. You should write all standards used in the analysis.
7. More recent references should be added to the introduction such as :
a) Numerical analysis of the shear behavior of FRP-strengthened continuous RC beams having web openings.
b) Behavior of RC beams strengthened in shear with ultra-high performance fiber reinforced concrete (UHPFRC).
c) Behavior of steel I-beam embedded in normal and steel fiber reinforced concrete incorporating demountable bolted connectors.
d) Bond behavior between concrete and prefabricated Ultra High-Performance Fiber-Reinforced Concrete (UHPFRC) plates.
e) Effect of interfacial surface preparation technique on bond characteristics of both NSC-UHPFRC and NSC-NSC composites.
f) Flexural strengthening of RC one way solid slab with Strain Hardening Cementitious Composites (SHCC).
g) Finite element analysis of shear performance of UHPFRC-encased steel composite beams: Parametric study
10. How can the results of this paper help to improve the specification?
11. The research significance of the current study should be added to the introduction.
12. Quality of all figures should be enhanced
Round 2
Reviewer 2 Report
To improve the quality of this manuscript, some figures need major revision. The precision of Figs. 7, 9, 10, 13 - 16 are not clear to the reviewer.
Author Response
Noted.
The figures underwent additional improvements; I hope now they are more to your liking.
Reviewer 3 Report
The authors adequately revised the paper.
Author Response
Thank you.